# A simple method for detecting chaos in nature

Daniel Toker ⬛ [1]*, Friedrich T. Sommer[1] & Mark D'Esposito[1]

Chaos, or exponential sensitivity to small perturbations, appears everywhere in nature. Moreover, chaos is predicted to play diverse functional roles in living systems. A method for detecting chaos from empirical measurements should therefore be a key component of the biologist's toolkit. But, classic chaos-detection tools are highly sensitive to measurement noise and break down for common edge cases, making it difficult to detect chaos in domains, like biology, where measurements are noisy. However, newer tools promise to overcome these limitations. Here, we combine several such tools into an automated processing pipeline, and show that our pipeline can detect the presence (or absence) of chaos in noisy recordings, even for difficult edge cases. As a first-pass application of our pipeline, we show that heart rate variability is not chaotic as some have proposed, and instead reflects a stochastic process in both health and disease. Our tool is easy-to-use and freely available.

---

[1] Helen Wills Neuroscience Institute, University of California, Berkeley, CA, USA. *email: danieltoker@berkeley.edu

A remarkable diversity of natural phenomena are thought to be chaotic. Formally, a system is chaotic if it is bounded (meaning that, like a planet circling a star, its dynamics stay inside an orbit rather than escaping off to infinity), and if it is deterministic (meaning that, with the exact same initial conditions, it will always evolve over time in the same way), and if tiny perturbations to the system get exponentially amplified (Glossary (Supplementary Information), Supplementary Figs. 1, 2). The meteorologist Edward Lorenz famously described this phenomenon as the butterfly effect: in a chaotic system, something as small as the flapping of a butterfly's wings can cause an effect as big as a tornado. This conceptually simple phenomenon—i.e., extreme sensitivity to small perturbations—is thought to appear everywhere in nature, from cosmic inflation[1], to the orbit of Hyperion[2], to the Belousov–Zhabotinskii chemical reaction[3], to the electrodynamics of the stimulated squid giant axon[4]. These are only a few examples of the many places in nature where chaos has been found.

It is relatively simple to determine if a simulated system is chaotic: just run the simulation a few times, with very slightly different initial conditions, and see how quickly the simulations diverge (Supplementary Fig. 1). But, if all that is available are measurements of how a real, non-simulated system evolves over time—for e.g., how a neuron's membrane potential changes over time, or how the brightness of a star changes over time— how can it be determined if those observations come from a chaotic system? Or if they are just noise? Or if the system is in fact periodic (Glossary, Supplementary Figs. 1, 2), meaning that, like a clock, small perturbations do not appreciably influence its dynamics?

While a reliable method for detecting chaos using empirical recordings should be an essential part of any scientist's toolbox, such a tool might be especially helpful to biologists, as chaos is predicted to play an important functional role in a wide variety of biological processes (that said, we note that real biological systems cannot be purely chaotic in the strict mathematical since, since they certainly contain some level of dynamic noise—see Glossary —but that researchers have long speculated that many biological processes are still predominantly deterministic, but chaotic[5]). For example, following early speculations about the presence of chaos in the electrodynamics of both cardiac[6] and neural[7] tissue, the science writer Robert Pool posited in 1989 that "chaos may provide a healthy flexibility for the heart, brain, and other parts of the body."[8] Though this point has been intensely debated since the 1980s[5,9], a range of more specific possible biological functions for chaos have since been proposed, including potentially maximizing the information processing capacity of both neural systems[10] and gene regulatory networks[11], enabling multistable perception[12], allowing neural systems to flexibly transition between different activity patterns[13], and boosting cellular survival rates through the promotion of heterogeneous gene expression[14]. And there is good reason to expect chaos to exist in biological systems, as a large range of simulations of biological processes[15], and in particular of neural systems[9], show clear evidence of chaos. Moreover, unambiguous evidence of biological chaos has been found in a very small number of real cases that were amenable to comparison to good theoretical models; these include periodically stimulated squid giant axons[4] and cardiac cells[16], as well as the discharges of the *Onchidium* pacemaker neuron[17] and the *Nitella flexillis* internodal cell[18]. But, beyond simulations and these select empirical cases, most attempts to test the presence or predicted functions of chaos in biology have fallen short due to high levels of measurement noise (Glossary) in biological recordings. For this reason, it has long been recognized that biologists need a noise-robust tool for detecting the presence (or absence) of chaos in their noisy empirical data[9,15].

Researchers also need a tool that can detect varying degrees of chaos (Glossary) in noisy recordings. In strongly chaotic systems, initially similar system states diverge faster than they do in weakly chaotic systems. And such varying degrees of chaos are predicted to occur in biology, with functional consequences. For example, a model of white blood cell concentrations in chronic granulocytic leukemia can display varying levels of chaos, and knowing how chaotic those concentrations are in actual leukemia patients could have important implications for health outcomes[19]. As another example, models of the human cortex predict that macro-scale cortical electrodynamics should be weakly chaotic during waking states and should be strongly chaotic under propofol anesthesia[20]; if this prediction is true, then detecting changing levels of chaos in large-scale brain activity could be useful for monitoring depth of anesthesia and for basic anesthesia research. Thus, it is imperative to develop tools that can not only determine that an experimental system is chaotic, but also tools to assess changing levels of chaos in a system.

Although classic tools for detecting the presence and degree of chaos in data are slow, require large amounts of data, are highly sensitive to measurement noise, and break down for common edge cases, more recent mathematical research has provided new, more robust tools for detecting chaos or a lack thereof in noisy time-series recordings. Here, for the first time (to our knowledge), we combine several key mathematical tools into a single, fully automated Matlab processing pipeline, which we call the Chaos Decision Tree Algorithm[21] (Fig. 1). The Chaos Decision Tree Algorithm takes a single time-series of any type—be it recordings of neural spikes, time-varying transcription levels of a particular gene, fluctuating oil prices, or recordings of stellar flux - and classifies those recorded data as coming from a system that is predominantly (or "operationally"[22]) stochastic, periodic, or chaotic. The algorithm requires no input from the user other than a time-series recording, though we have structured our code such that users can also select from among a number of alternative subroutines (see Methods section, Fig. 1).

In this paper, we show that the Chaos Decision Tree Algorithm performs with very high accuracy across a wide variety of both real and simulated systems, even in the presence of relatively high levels of measurement noise. Moreover, our pipeline can accurately track changing degrees of chaos (for e.g., transitions from weak to strong chaos). With an eye toward applications to biology, the simulated systems we tested included a high-dimensional mean-field model of cortical electrodynamics, a model of a spiking neuron, a model of white blood cell concentrations in chronic granulocytic leukemia, and a model of the transcription of the NF-κB protein complex. We also tested the algorithm on a wide variety of non-biological simulations, including several difficult edge cases; these included strange non-chaotic systems, quasi-periodic systems, colored noise, and nonlinear stochastic systems (see Glossary for definitions of these terms), which are all classically difficult to distinguish from chaotic systems[23–26]. We also tested the algorithm on a hyperchaotic system (Glossary), which can be difficult to distinguish from noise[25], as well as on several non-stationary processes (Glossary) in order to test the robustness of the algorithm against non-stationarity. Finally, we tested the Chaos Decision Tree Algorithm on several empirical (i.e. non-simulated) datasets for which the ground-truth presence or absence of chaos has been previously established by other studies. These included an electronic circuit in periodic, strange non-chaotic, and chaotic states[27], a chaotic laser[28], the stellar flux of a strange non-chaotic star[29], the linear/stochastic North Atlantic Oscillation index[30], and nonlinear/stochastic Parkinson's and essential tremors[26]. Overall, our pipeline performed with near-perfect accuracy in classifying these data as stochastic, periodic, or chaotic, as well as

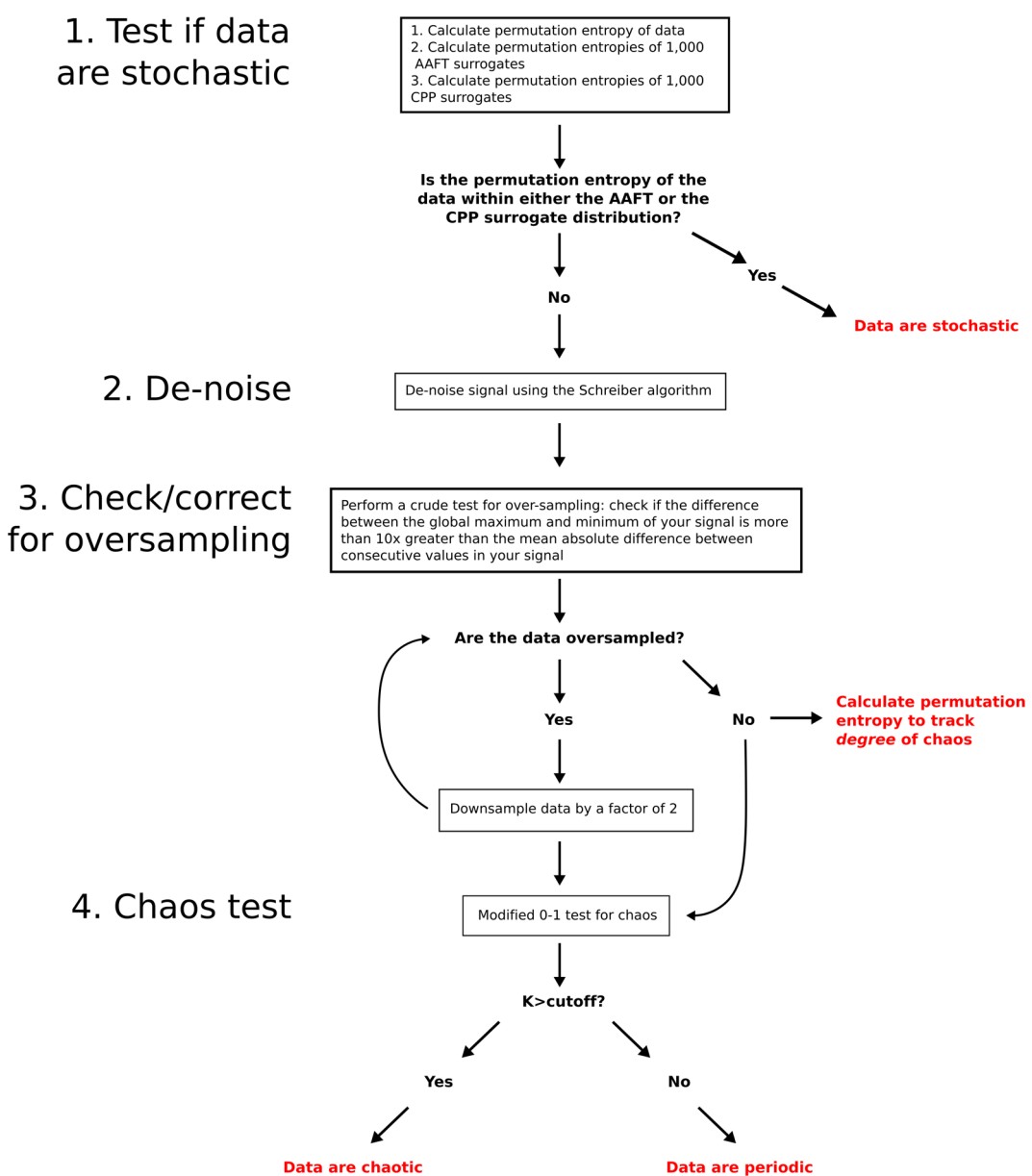

**Fig. 1 The Chaos Decision Tree Algorithm[21].** The first step of the algorithm is to test if data are stochastic. The Chaos Decision Tree Algorithm uses a surrogate-based approach to test for stochasticity, by comparing the permutation entropy of the original time-series to the permutation entropies of random surrogates of that time series. If the user does not specify which surrogate algorithms to use, the Chaos Decision Tree Algorithm automatically picks a combination of Amplitude Adjusted Fourier Transform[33] surrogates and Cyclic Phase Permutation[35] surrogates—see Supplementary Tables 2, 3. If the permutation entropy of the original time-series falls within either surrogate distribution, the time-series is classified as stochastic; if the permutation entropy falls outside the surrogate distributions, then the algorithm proceeds to denoise the inputted signal. Several de-noising subroutines are available, but if the user does not specify a subroutine, the pipeline will use Schreiber's noise-reduction algorithm[36] (Supplementary Table 5). The pipeline then checks for signal oversampling; if data are oversampled, the pipeline iteratively downsamples the data until they are no longer oversampled (note that an alternative downsampling method proposed by Eyébé Fouda and colleagues[68] may be selected instead—see Supplementary Table 6). Finally, the Chaos Decision Tree Algorithm performs the 0–1 chaos test on the input data, which has been modified from the original 0–1 test to be less sensitive to noise, suppress correlations arising from quasi-periodicity, and normalize the standard deviation of the test signal (see Methods section). The value for the parameter that suppresses signal correlations can be specified by the user, but is otherwise automatically chosen based on ROC analyses performed here (Supplementary Fig. 4). The modified 0–1 test provides a single statistic, $K$, which approaches 1 for chaotic systems and approaches 0 for periodic systems. Any cutoff for $K$ may be inputted to the pipeline, and if no cutoff is provided, the pipeline will use a cutoff based on the length of the time-series (Supplementary Fig. 6). If $K$ is greater than the cutoff, the data are classified as chaotic, and if they are less than or equal to the cutoff, they are classified as periodic.

in tracking changing degrees of chaos in both real and simulated systems. Finally, we applied our algorithm to electrocardiogram recordings from healthy subjects, patients with congestive heart failure, and patients with atrial fibrillation[31], and provide

evidence that heart rate variability reflects a predominantly stochastic, rather than chaotic process.

We have made our Matlab code freely and publicly available at https://figshare.com/s/80891dfb34c6ee9c8b34.

## Results

The Chaos Decision Tree Algorithm[21] is depicted graphically in Fig. 1. The pipeline consists of four steps: (1) Determine if the data are stochastic using permutation entropy[32] and a combination of Amplitude Adjusted Fourier transform surrogates[33,34] and Cyclic Phase Permutation surrogates[34,35] (Glossary), (2) Denoise the data using the Schreiber de-noising algorithm[36] (Glossary), (3) Correct for possible signal oversampling, and (4) Test for chaos using a modified 0–1 test for chaos[23,37–40] (Glossary). For each step of the processing pipeline, we compared the performance of different available tools (i.e., different surrogate-based tests for stochasticity, different de-noising methods, different downsampling methods, and different chaos-detection methods), and chose the tools with the highest classification performance (Supplementary Tables 1–14). Note that with user input, the Chaos Decision Tree Algorithm can use any of the alternative tools tested here, and that with no user input other than a time-series recording, the algorithm will automatically use the tools we found maximized its performance. All results reported in the main body of this paper are for this automated set of high-performing tools. See Supplementary Fig. 3 for example time-traces illustrating each step of the algorithm.

We tested the performance of the (automated) Chaos Decision Tree Algorithm in detecting the presence and degree of chaos in a wide range of simulated and empirical systems for which the ground-truth presence of chaos, periodicity, or stochasticity has already been established. Details about each dataset and how the ground-truth presence or absence of chaos in those systems was previously determined are included in the Methods. Note that some systems are labeled "SNA," which is an abbreviation for "strange non-chaotic attractor" (Glossary). These are systems whose attractors in phase space (Glossary) are fractal (like chaotic systems), but which are periodic (i.e., non-chaotic). Among these, we included the only known non-artificial strange non-chaotic system, the stellar flux of the so-called golden star KIC 5520878, as recorded by the Kepler space telescope[29]. All simulated datasets consisted of 10,000 time-points, and all initial conditions were randomized. For systems with more than one variable, we here report results for linear combinations of those variables (see Methods section), under the assumption that in most real-life cases, empirical recordings will contain features of multiple components of the system of study; that said, we also confirmed that the Chaos Decision Tree Algorithm has very high performance for individual system variables (Supplementary Table 15).

Results for simulations of biological systems are reported in Table 1, and results for non-biological simulations are reported in Table 2. Note that no measurement noise was added to the colored noise signals in Table 2, as doing so would flatten their power spectra. Because the datasets in Tables 1 and 2 were used to choose between alternative methods for detecting stochasticity (Supplementary Tables 1–4), de-noising (Supplementary Table 5), downsampling (Supplementary Table 6), and alternative

tests of chaos (Supplementary Tables 8–13), as well as to optimize the 0–1 test for chaos (Supplementary Figs. 4–6), we further tested the Chaos Decision Tree Algorithm on held out datasets, which were not used to adjudicate between alternative tools. These held out datasets included both simulated systems (Table 3) and recordings from real (non-simulated) systems (Table 4). Several of these held out datasets were of direct biological relevance: the periodically stimulated Poincaré oscillator in Table 3 is thought to be a good model of cardiac cell electrodynamics[41], which, like the Poincaré oscillator, are chaotic when periodically stimulated with certain delays between stimulation pulses[16]; the integrated circuit in Table 4 is a physical implementation of equations that are based on the Hodgkin-Huxley neuron model[42]; and the tremor signals in Table 4 are direct recordings from patients. The Chaos Decision Tree Algorithm classified the systems in Tables 1–4 as stochastic, periodic, or chaotic with near-perfect accuracy even at high levels of measurement noise, with the exception of the noise-driven sine map (Table 2)—see Discussion section. Finally, we tested the performance of the Chaos Decision Tree Algorithm on sub-samples of all systems in Tables 1–3, and confirmed that it is still highly accurate for data with just 1000 time-points (Supplementary Table 16) or 5000 time-points (Supplementary Table 17), though we note that performance for some systems did go down with less data, which is to be expected[40].

Table 5 reports the accuracy of the Chaos Decision Tree Algorithm in detecting degree of chaos. Formally, a system's degree of chaos is quantified by the magnitude of its largest Lyapunov exponent (Glossary). Unfortunately, largest Lyapunov exponents are very difficult to estimate from finite, noisy time-series recordings. But, directly estimating largest Lyapunov exponents may not be necessary for tracking changing degrees of chaos in real systems: following prior observations of a strong correlation between a quick-to-compute and noise-robust measure called permutation entropy (Glossary) and the largest Lyapunov exponents of several systems[32,43], the Chaos Decision Tree Algorithm approximates degree of chaos by calculating the permutation entropy of the inputted signal, after it has been de-noised and corrected for possible over-sampling. In agreement with prior findings, we found that permutation entropy tracked degree of chaos in the logistic map, the Hénon map, the Lorenz system, a high-dimensional mean-field model of the cortex, and an electronic circuit. See Methods for details on the parameters that were used to generate dynamics with different degrees of chaos in these systems, and for details on how ground-truth largest Lyapunov exponents were calculated. Note that without downsampling, the correlation between the largest Lyapunov exponents and permutation entropy breaks down in continuous systems (Supplementary Table 14), which is to be expected, as permutation entropy has only been analytically proven to track the degree of chaos in discrete-time systems[32,44] (see Glossary).

**Table 1 The Chaos Decision Tree Algorithm classified biological simulations as periodic or chaotic with near-perfect accuracy.**

| System | Measurement noise level (% of std. dev.) | | | | |
|---|---|---|---|---|---|
| | 0% | 10% | 20% | 30% | 40% |
| Cortical model[20] (chaotic) | 100/100 | 100/100 | 100/100 | 97/100 | 83/100 |
| Cortical model[20] (periodic) | 100/100 | 100/100 | 100/100 | 100/100 | 100/100 |
| Spiking neuron[49] (chaotic) | 100/100 | 100/100 | 100/100 | 100/100 | 98/100 |
| Granulocyte levels[19] (chaotic) | 100/100 | 100/100 | 100/100 | 100/100 | 100/100 |
| Granulocyte levels[19] (periodic) | 100/100 | 100/100 | 100/100 | 100/100 | 83/100 |
| NF-$\kappa$B transcription[14] (chaotic) | 99/100 | 99/100 | 100/100 | 100/100 | 100/100 |
| NF-$\kappa$B transcription[14] (periodic) | 100/100 | 100/100 | 97/100 | 100/100 | 100/100 |

**Table 2 The Chaos Decision Tree Algorithm classified non-biological simulations as stochastic, periodic, or chaotic with high accuracy. These simulated systems include strange non-chaotic attractors (SNAs), linear stochastic processes, and nonlinear stochastic processes, all of which are classically difficult to distinguish from chaos.**

| | Measurement noise level (% of std. dev.) | | | | |
| --- | --- | --- | --- | --- | --- |
| System | 0% | 10% | 20% | 30% | 40% |
| Cubic map[50] (chaotic) | 100/100 | 100/100 | 100/100 | 100/100 | 100/100 |
| Cubic map[50] (periodic) | 100/100 | 100/100 | 100/100 | 100/100 | 100/100 |
| Cubic map[50] (SNA HH) | 100/100 | 100/100 | 100/100 | 100/100 | 100/100 |
| Cubic map[50] (SNA S3) | 100/100 | 100/100 | 100/100 | 100/100 | 0/100 |
| GOPY map[51] (SNA) | 100/100 | 100/100 | 100/100 | 99/100 | 14/100 |
| Logistic map[52] (chaotic) | 100/100 | 100/100 | 100/100 | 100/100 | 100/100 |
| Logistic map[52] (periodic) | 100/100 | 100/100 | 100/100 | 100/100 | 100/100 |
| Lorenz system[53] (chaotic) | 100/100 | 100/100 | 97/100 | 82/100 | 36/100 |
| Generalized Hénon map[54] (hyperchaotic) | 100/100 | 100/100 | 100/100 | 100/100 | 93/100 |
| Freitas map[55] (nonlinear stochastic) | 78/100 | 83/100 | 98/100 | 98/100 | 74/100 |
| Noise-driven sine map[55] (nonlinear stochastic) | 55/100 | 3/100 | 22/100 | 5/100 | 78/100 |
| Bounded random walk[56] (nonlinear stochastic) | 100/100 | 97/100 | 59/100 | 95/100 | 100/100 |
| Cyclostationary process[57] (linear stochastic) | 99/100 | 100/100 | 99/100 | 100/100 | 100/100 |
| ARMA process (linear stochastic) | 85/100 | 98/100 | 99/100 | 99/100 | 100/100 |
| Trended random walk (linear stochastic) | 100/100 | 89/100 | 90/100 | 98/100 | 100/100 |
| Random walk (linear stochastic) | 100/100 | 98/100 | 100/100 | 100/100 | 100/100 |
| Violet noise[58] (linear stochastic) | 99/100 | | | | |
| Blue noise[58] (linear stochastic) | 100/100 | | | | |
| White noise[58] (linear stochastic) | 100/100 | | | | |
| Pink noise[58] (linear stochastic) | 100/100 | | | | |
| Red noise[58] (linear stochastic) | 100/100 | | | | |

**Table 3 Classification accuracy in held out simulated systems. While the datasets in Tables 1, 2 were used to optimize the Chaos Decision Tree Algorithm, these datasets were not. Performance was near-perfect.**

| | Measurement noise level (% of std. dev.) | | | | |
| --- | --- | --- | --- | --- | --- |
| System | 0% | 10% | 20% | 30% | 40% |
| Rössler system[59] (chaotic) | 70/100 | 90/100 | 96/100 | 100/100 | 100/100 |
| Ikeda map[60] (chaotic) | 100/100 | 100/100 | 100/100 | 100/100 | 14/100 |
| Hénon map[61] (periodic) | 100/100 | 100/100 | 100/100 | 100/100 | 100/100 |
| Cubic map[61] (period-doubled) | 100/100 | 100/100 | 100/100 | 100/100 | 100/100 |
| Poincaré oscillator[41] (periodic) | 100/100 | 100/100 | 100/100 | 100/100 | 57/100 |
| Poincaré oscillator[41] (quasi-periodic) | 100/100 | 100/100 | 100/100 | 100/100 | 100/100 |
| Poincaré oscillator[41] (chaotic) | 100/100 | 100/100 | 100/100 | 100/100 | 100/100 |
| Multivariate AR model (linear stochastic) | 100/100 | 100/100 | 100/100 | 100/100 | 100/100 |

Finally, as a first-pass implementation of our method, we applied the Chaos Decision Tree Algorithm to recordings of human heart rate variability, made available by Physionet[31]. There has been considerable debate over whether or not the irregularities of heart rate signals (in either health or disease) reflect a predominantly chaotic process. While many classic chaos-detection methods have identified heart rate variability as chaotic (see Glass[5] for a review), other studies have argued that this is an erroneous classification, suggesting that heart rate variability is, in fact, a nonlinear stochastic process[45,46], and that prior classifications of heart rate signals as chaotic simply reflect the shortcomings of classic chaos-detection methods. In agreement with this view, we here show that the Chaos Decision Tree Algorithm classified heart rate signals from healthy subjects, congestive heart failure patients, and atrial fibrillation patients as stochastic, rather than chaotic, with the exception of two congestive heart failure patients (Table 6).

**Discussion**

In this paper, we have introduced a processing pipeline, called the Chaos Decision Tree Algorithm[21], that can accurately detect whether a time-series signal is generated by a predominantly stochastic, periodic, or chaotic system, and can also accurately track changing levels of chaos within a system using permutation entropy. The pipeline makes no assumptions about the input data. The Chaos Decision Tree Algorithm consists of four broad steps: (1) testing for stochasticity using surrogate data methods, (2) de-noising, (3) downsampling if data are oversampled, and (3) testing for chaos using the modified 0–1 test. We tested the performance of several different surrogate data generation algorithms, de-noising algorithms, downsampling algorithms, and parameters for the modified 0–1 test. Each alternative algorithm and parameter choice has its relative strengths and weaknesses, and we have structured our code such that a user can specify which algorithms and parameters to use for each step of the pipeline. If a user only inputs a time-series recording without specifying any sub-algorithms or parameters, then our pipeline will automatically use the methods and parameters we found yielded the most accurate results across a large and diverse set of data. All analyses reported in the main body of this paper are for this automated set of subroutines.

We tested the (automated) Chaos Decision Tree Algorithm on a diverse range of simulations of biological systems, non-

biological simulations, and empirical (non-simulated) data recordings. Empirical data were recorded from an integrated circuit in a periodic, strange non-chaotic, and chaotic state, a chaotic laser, the stellar flux of a strange non-chaotic star, the North Atlantic Oscillation index, a Parkinson's tremor, an essential tremor, and heart rate variability from congestive heart failure patients, atrial fibrillation patients, and healthy controls. In the cases for which the ground-truth was known (i.e., all datasets other than heart rate variability), the Chaos Decision Tree Algorithm performed at very high accuracy even at relatively high levels of measurement noise. For heart rate variability, our results support the hypothesis that cardiac rhythm variability is stochastic[45,46]. Overall, these findings make us confident that the Chaos Decision Tree Algorithm can be fruitfully applied to biological and non-biological signals contaminated by measurement noise.

We note a few limitations/shortcomings of our algorithm. First, the 0–1 test used in our pipeline might classify some very weakly chaotic systems (i.e., systems whose largest Lyapunov exponent is positive but very near zero) as periodic if the length of the time-series provided is short; but, with longer time-series, the test is guaranteed to provide accurate results[40]. We also note that the algorithm performed poorly for the noise-driven sine map, which was consistently mis-classified as chaotic (Table 2). It is possible that this system was not classified as stochastic because its level of intrinsic noise was very low; in support of this, we found that the Chaos Decision Tree Algorithm classified nonlinear dynamical systems with very low levels of intrinsic noise as deterministic, and that classifications of stochasticity became more frequent as

the level of intrinsic noise was increased (Supplementary Table 18). It is also possible that this system is, in fact, an example of noise-induced chaos[47]. Finally, although the choice of system observables did not appreciably affect the performance of our method (Supplementary Table 15), we agree with Letellier and colleagues[48] that some system observables are better representations of a system's dynamics than others, and that this can have important consequences for the accuracy of nonlinear time-series analysis methods such as this one. In light of these potential limitations, it bears re-emphasizing that the absence, presence, and degree of chaos can only be determined with absolute certainty in a computer model that is free of measurement noise, by running multiple simulations and seeing how quickly initially similar states diverge. Thus, although the Chaos Decision Tree Algorithm pipeline performs at very high accuracy, it should, when possible, be used in conjunction with analyses of a computer model of the system at hand.

We hope that the Chaos Decision Tree Algorithm will help advance the decades-old effort to bring the insights of chaos theory to biology. While a diverse range of biological simulations and a small number of real biological cases have been shown to be chaotic, detecting the presence and degree of chaos in biological recordings has been difficult. The Chaos Decision Tree Algorithm overcomes the difficulties of prior tests, by being fast, highly robust to measurement noise, and, unless the user specifies specific alternative subroutines, fully automated. We welcome any efforts to identify edge cases for which our pipeline systematically breaks down; given that our pipeline is a modular decision tree, new subroutines can be added to accommodate such edge cases. We hope that our pipeline (and perhaps future iterations of it) will be useful to any of the domains of science—and in particular of biology—in which chaos has been invoked, but not tested.

## Methods

**The Chaos Decision Tree Algorithm**. To understand the logic of the Chaos Decision Tree Algorithm[21], we begin with the final test in the decision tree. The crux of the Chaos Decision Tree Algorithm is the 0–1 test for chaos. The 0–1 test for chaos was originally developed by Gottwald and Melbourne[37], who later offered a slightly modified version of the test, which can cope with moderate amounts of measurement noise[38]. Several years later, Dawes and Freeland further modified the test, such that it could suppress correlations induced by quasi-periodic dynamics, and thus more effectively distinguish between chaotic and strange non-chaotic dynamics, which are difficult to distinguish given only a time-series recording[23]. The modified 0–1 test involves taking a one-dimensional time-series of interest $\phi$, and using it to drive the following two-dimensional system:

$$
\begin{aligned}
p(n+1) &= p(n) + \phi(n) \cos cn \\
q(n+1) &= q(n) + \phi(n) \sin cn
\end{aligned}
\tag{1}
$$

where $c$ is a random value bounded between 0 and $2\pi$. For a given $c$, the solution to

---

**Table 4 Classification accuracy in empirical (non-simulated) data from stochastic, periodic, strange non-chaotic (SNA), and chaotic systems. As was the case for the data in Table 3, these datasets were not used to optimize algorithm performance, and so are also held out datasets. Algorithm performance was perfect.**

| System | |
| --- | --- |
| Neuron integrated circuit[27] (chaotic) | 10/10 |
| Neuron integrated circuit[27] (SNA) | 10/10 |
| Neuron integrated circuit[27] (periodic) | 10/10 |
| Laser[28] (chaotic) | 1/1 |
| Stellar flux[29] (SNA) | 1/1 |
| North Atlantic Oscillation index[30] (linear stochastic) | 1/1 |
| Essential tremor[26] (nonlinear stochastic) | 1/1 |
| Parkinson's tremor[26] (nonlinear stochastic) | 1/1 |

---

**Table 5 The Chaos Decision Tree Algorithm uses permutation entropy, calculated from data that have been de-noised and downsampled (if oversampled), to track the degree of chaos in a system, which might change as the state of the system changes.**

| System | Measurement noise level (% of std. dev.) | | | | |
| --- | --- | --- | --- | --- | --- |
| | 0% | 10% | 20% | 30% | 40% |
| Logistic map[52] | 0.93*** | 0.80*** | 0.93*** | 0.93*** | 0.91*** |
| Hénon map[61] | 0.92*** | 0.93*** | 0.94*** | 0.92*** | 0.88*** |
| Lorenz system[53] | 0.81*** | 0.71*** | 0.73*** | 0.69*** | 0.62*** |
| Cortical model[20] | 0.69*** | 0.55*** | 0.39*** | 0.33*** | 0.24*** |
| Neuron integrated circuit[27] | 0.94*** | | | | |

We here show Spearman correlations between permutation entropy and largest Lyapunov exponents, which measure degree of chaos but which are difficult to estimate from empirical data. Data include four simulated systems and recordings from an integrated circuit in different states. See Methods for how ground-truth largest Lyapunov exponents were calculated for these systems
***$p < 0.001$ (two-tailed) after Bonferroni-correcting for multiple comparisons to the same set of ground-truth largest Lyapunov exponents

**Table 6 The Chaos Decision Tree Algorithm consistently classifies heart rate recordings, across conditions, as stochastic.**

| Condition | Classification | | |
|---|---|---|---|
| | Stochastic | Periodic | Chaotic |
| Healthy controls[31] | 5/5 | 0/5 | 0/5 |
| Congestive heart failure[31] | 3/5 | 2/5 | 0/5 |
| Atrial fibrillation[31] | 5/5 | 0/5 | 0/5 |

The only exceptions were the heart rate signals recorded from two patients with congestive heart failure, which were classified as periodic

Eq. (1) yields:

$$p_c(n) = \sum_{j=1}^{n} \phi(j) \cos jc$$

$$q_c(n) = \sum_{j=1}^{n} \phi(j) \sin jc \qquad (2)$$

Gottwald and Melbourne show that if the inputted time-series $\phi$ is regular, the motion of **p** and **q** is bounded, while **p** and **q** display asymptotic Brownian motion if $\phi$ is chaotic. The time-averaged mean square displacement of **p** and **q**, plus the noise term proposed by Dawes and Freeland[23], is

$$M_c(n) = \frac{1}{N} \sum_{j=1}^{N} ([p_c(j+n) - p_c(j)]^2 + [q_c(j+n) - q_c(j)]^2) + \sigma \eta_n. \qquad (3)$$

where $\eta_n$ is a uniformly distributed random variable between $\left[-\frac{1}{2}, \frac{1}{2}\right]$ and $\sigma$ is the noise level. Finally, the outputted $K$-statistic of the 0–1 test uses a correlation coefficient to measure the growth rate of the mean squared displacement of the two-dimensional system in Eq. (1):

$$K_c = \mathrm{corr}(n, M_c(n)) \qquad (4)$$

$K$ is computed for 100 different values of $c$, randomly sampled between 0 and $2\pi$, and the final output of the test is the median $K$ across different values of $c$. For chaotic systems, this median $K$ value will approach 1, and for periodic systems, $K$ will approach 0[23,37–40].

There are two parameters in this modified 0–1 test: the parameter $\sigma$ that controls the level of added noise in Eq. (3), and the cutoff for what $K$-statistic values are classified as indicating chaos or periodicity in a finite time-series. We performed ROC-curve analyses for different values of $\sigma$ and found that $\sigma = 0.5$ maximized classification performance across systems and noise levels (Supplementary Fig. 4), and so our pipeline automatically sets $\sigma$ to 0.5 if $\sigma$ is not specified by the user. Note that for non-zero values of $\sigma$, $K$ approaches zero as the standard deviation of a test signal approaches zero (Supplementary Fig. 5), and so the Chaos Decision Tree Algorithm multiplies a test signal by a constant to fix its standard deviation at 0.5 before applying the 0–1 test. A cutoff for $K$ can also be inputted to our Matlab script, such that data that yield a $K$ value greater than that cutoff are classified as chaotic and data that yield a $K$ value less than or equal to that cutoff are classified as periodic. If no cutoff is provided, a cutoff is chosen based on an analysis of optimal cutoffs as a function of time-series length (Supplementary Fig. 6). If the automatically selected cutoff is greater than 0.99, the cutoff is set to $K = 0.99$, as $K$ is upper-bounded by 1. We have confirmed that this automated cutoff selection yields highly accurate results for sub-samples of both test and held-out datasets (Supplementary Tables 16, 17).

The 0–1 test described above only yields accurate results for data that are deterministic[24,40,62,63]. A system is considered deterministic if, given the exact same initial conditions, it always evolves over time the same way, whereas a system is considered stochastic if there is appreciable randomness built in to its evolution over time (Glossary, Supplementary Fig. 1, 2). Not only are all chaotic systems (predominantly) deterministic—and thus the possibility of chaos can be automatically rejected if a system is found to be stochastic (though we note that a mathematically rigorous definition of chaos has recently been extended to the domain of stochastic systems, under the framework of the Supersymmetric Theory of Stochastics[47])—but the 0–1 test is also known to incorrectly classify stochastic dynamics as chaotic[24,62,63]. Thus, the Chaos Decision Tree Algorithm first rules out the possibility that data are predominantly stochastic before applying the modified 0–1 test. To do so, it uses a noise-robust method recently developed by Zunino and Kulp[64], which tests for determinism using surrogate statistics[33], with permutation entropy[32] as the test statistic. The calculation of permutation entropy relies on two parameters: permutation order and time-lag. We follow the recommendation from Bandt and Pompe[32] and set the time-lag to 1, and found that a permutation order of 8 maximized stochasticity detection accuracy (Supplementary Tables 2, 3). Moreover, we use a combination of amplitude adjusted Fourier transform

surrogates[33] and Cyclic Phase Permutation surrogates[35], unlike Zunino and Kulp, who used iterative amplitude adjusted Fourier transform[33] surrogates, because we found that this combination led to far higher classification accuracy (Supplementary Tables 2, 3). The Chaos Decision Tree Algorithm classifies data as stochastic (and thus does not proceed to subsequent steps) if the permutation entropy of the original data falls within either surrogate distribution. The algorithm uses the Toolboxes for Complex Systems implementation of the permutation entropy algorithm, written by Andreas Müller[65]. Surrogates are generated using the Matlab toolbox recently released by Lancaster and colleagues[34]. Note that because Fourier-based surrogates are strictly stationary, surrogate-based tests that use only Fourier-based algorithms are only valid if the test time-series is also stationary[34,57]; that said, we found that non-stationarity did not affect the accuracy of a stochasticity test that uses a combination of amplitude adjusted Fourier transform and Cyclic Phase Permutation surrogates (Supplementary Tables 1–4). We also did not find that a normality transformation of the data improved the performance of our surrogate-based stochasticity test (Supplementary Table 2), counter to what has been suggested elsewhere[22].

If data "pass" the stochasticity test described above and are deemed operationally deterministic, then the Chaos Decision Tree Algorithm automatically denoises the inputted signal. We compared three de-noising algorithms: a moving average filter (using Matlab's smooth.m function), the Matlab Chaotic Systems Toolbox's[66] implementation of Schreiber's noise-reduction algorithm[36] (Glossary), and wavelet de-noising using an empirical Bayesian method with a Cauchy prior (using Matlab's wdenoise.m function). Although it is considerably slower to run, Schreiber denoising markedly outperforms the other two approaches in recovering the deterministic component of signals contaminated by measurement noise (Supplementary Table 5), and markedly improves the performance of the modified 0–1 test (Supplementary Table 6, Supplementary Fig. 4). Thus, the Chaos Decision Tree Algorithm automatically uses Schreiber de-noising before testing for chaos, unless the user specifies one of the other two de-noising algorithms tested here to be used instead.

The final step of the Chaos Decision Tree Algorithm before applying the 0–1 test is to check if data are oversampled and to downsample them if they are. Gottwald and Melbourne have shown[39] that the 0–1 test can give inaccurate results for continuous (i.e., non-discrete-time) systems sampled at a very high frequency, but that it can accurately differentiate between periodic dynamics and chaotic dynamics in continuous deterministic systems when data are properly downsampled. In light of this, the Chaos Decision Tree Algorithm utilizes the (crude) test for oversampling used by Matthews[67], by calculating a measure $\eta$, which is the difference between the global maximum and global minimum of the data divided by the mean absolute difference between consecutive time-points in the data. If $\eta > 10$, then the data are deemed to be oversampled, and the Chaos Decision Tree Algorithm iteratively downsamples the data by a factor of 2 until $\eta \le 10$ or until there are fewer than 100 time-points left in the signal. We compared this approach both to no downsampling and to an alternative method, suggested by Eyébé Fouda and colleagues[68] to improve 0–1 test performance, which downsamples by taking just the local minima and maxima of oversampled signals. We found that downsampling after de-noising yields more accurate results than either alternative approach when oversampled signals are contaminated by measurement noise (Supplementary Table 6). We also note that recorded experimental data may be unlikely to be oversampled (Supplementary Table 7), and that this problem may be more likely to arise in simulated continuous systems. If the data are not oversampled, or if they have been downsampled, the Chaos Decision Tree Algorithm then applies the modified 0–1 test to the data, as described above.

Finally, the algorithm uses the permutation entropy of the inputted signal as a proxy for the degree of chaos in the system. Though the algorithm uses permutation entropy to establish whether or not a signal is predominantly deterministic (see above), permutation entropy has also been shown to tightly track the largest Lyapunov exponent (and therefore the degree of chaos) of the logistic map[32], the tent map[69], and the Duffing oscillator[43]. We should in general expect a close correspondence between permutation entropy and Lyapunov exponents, in light of the equivalence in discrete-time systems between permutation entropy and Kolmogorov-Sinai entropy[44,70–72], which is upper-bounded by the sum of a system's positive Lyapunov exponents—a relationship known as the "Pesin identity"[73]. When calculating permutation entropy to track degree of chaos (rather than for determinism testing as above), we follow Bandt and Pompe's[32] recommendation and simply set the time-lag to 1 and the permutation order to 5, which we showed tracks degree of chaos in all systems tested (Table 5). Because this equivalence is only known to hold for discrete-time systems[44], permutation entropy is only calculated after the inputted signal has been de-noised and, if oversampled, downsampled; this considerably improves its ability to track degree of chaos in continuous systems (Table 5, Supplementary Table 14).

The full decision tree of our algorithm is depicted graphically in Fig. 1.

## Data

*Biological simulations*. The following describes the simulations of biological systems analyzed in this paper. We only picked biological simulations for which the presence or absence of chaos has been established in prior work. Initial conditions were randomized in all simulations. We also tested the effect of measurement noise

on the accuracy of the Chaos Decision Tree Algorithm in classifying systems, by adding white noise to our simulated data, the amplitude of which was up to 40% the standard deviation of the original data. For each simulated system and level of measurement noise, we created 100 datasets with 10,000 time points.

*Chaotic mean-field cortical model.* Steyn-Ross, Steyn-Ross, and Sleigh[20] describe a mean-field model of the cortex based on the equations first introduced by Liley and colleagues[74,75], which includes electrical gap-junction synapses in addition to the standard chemical synapses used in the earlier models. The model contains both inhibitory and excitatory neural populations communicating locally through gap junctions and chemical synapses and communicating over long ranges via myelinated axons. The dynamics of each neural population in the model are determined by two first-order and six second-order partial differential equations, which is equivalent to 14 first-order differential equations. The primary output of the model is the mean excitatory firing rates of 120 neural populations, which approximates the large-scale cortical signals that might be measured through electrocortigraphy, magnetoencephalography, or electroencephalography. Steyn-Ross, Steyn-Ross, and Sleigh[20] show that by varying the inhibitory gap-junction diffusive-coupling strength parameter in their model, they can produce dynamics ranging from periodicity to strong chaos. In their simulation of "waking" cortical dynamics, Turing (spatial) and Hopf (temporal) instabilities interact to produce chaotic, low-frequency spatiotemporal oscillations. For chaotic dynamics, we simulated 2,000,000 time-points of their "wake" simulation, with the inhibitory gap-junction diffusive-coupling strength parameter set to 0.4, and then downsampled the data to 10,000 time-points. We only applied our algorithm to the mean excitatory firing rate of one neural population, i.e. to just one out of 14 variables describing the dynamics of just one out of 120 interacting such 14-dimensional systems (though the variable is biologically well-defined). The Matlab code for the simulations is available in the Supplementary Material of Steyn-Ross, Steyn-Ross, and Sleigh[20].

*Periodic mean-field cortical model.* Steyn-Ross, Steyn-Ross, and Sleigh show that their cortical mean-field model enters a periodic, seizure-like state dominated by a Hopf instability when the inhibitory gap-junction diffusive-coupling strength parameter is set to 0.1. Just as in the chaotic case, we simulated 2,000,000 time-points and then downsampled to 10,000 time-points. Note that Steyn-Ross, Steyn-Ross, and Sleigh estimate the largest Lyapunov exponent of their model to be around zero when the inhibitory gap-junction diffusive-coupling strength parameter is 0.1, whereas our own estimate (using an automated version of their same method—see below) placed the largest Lyapunov exponent more clearly in the periodic regime, at −2.1.

*Chaotic spiking neuron.* Izhikevich[49,76] describes a simple neuron model that can display both spiking and bursting behavior. The model consists of a neuron's membrane potential $v$, a membrane recovery variable $u$, an input current $I$, and parameters $a$, $b$, $c$, and $d$:

$$\frac{dv}{dt} = 0.004v^2 + 5v + 140 - u + I$$
$$\frac{du}{dt} = a(bv - u) \tag{5}$$

with the auxiliary after-spike resetting:

$$\text{if } v \geq +30\,\text{mV, then} \begin{cases} v \leftarrow c \\ u \leftarrow u + d. \end{cases} \tag{6}$$

When $a = 0.2$, $b = 2$, $c = -56$, $d = -16$, and $I = -99$, the neuron's membrane potential $v$ (which is the variable we analyze) displays chaotic spikes[49,76]. We simulated the Izhikevich neuron using a first-order Euler method, with an integration step of 0.25 ms. We generated 50,000 time points, and dowsampled by a factor of 5 (to avoid over-sampling).

*Periodic white blood cell concentration.* Inspired by the finding that chronic granulocytic leukemia involves apparently aperiodic oscillations in the concentration of circulating white blood cells[77], Mackey and Glass[19] study mathematical models of oscillating physiological control systems. They describe a simple mathematical model of the concentration of circulating white blood cells:

$$\frac{dx}{dt} = a\frac{x_\tau}{1 + x_\tau^c} - bx \tag{7}$$

where $a = 0.2$, $b = 0.1$, and $c = 10$. The parameter $\tau$ represents the delay between white blood cell production in bone marrow and the release of those cells into the blood stream. Since this cellular generation delay time is increased in some patients with chronic granulocytic leukemia, Mackey and Glass study the behavior of this system as a function of the delay time $\tau$. They find that as $\tau$ increased, the resulting oscillations produced by this equation became aperiodic. Through formal analysis of Lyapunov exponents of this system, Farmer[78] later confirmed that for $\tau = 10$, the oscillations of this system are periodic. We simulated 100,000 time-points of the periodic Mackey-Glass system using a first-order Euler method with an integration step of 1, and then downsampled by a factor of 10 (to avoid over-sampling).

*Chaotic white blood cell concentration.* For $\tau = 30$, Farmer confirmed[78] that the Mackey-Glass equation (Eq. (7)) for the concentration of circulating white blood cells yields a chaotic oscillation. We simulated 100,000 time-points of the chaotic

Mackey-Glass system using a first-order Euler method with an integration step of 1, and then downsampled by a factor of 10 (to avoid over-sampling).

*Periodic NF-κB transcription.* Heltberg and colleagues[14] recently described a five-dimensional mathematical model of oscillating concentrations of the transcription factor NF-κB, which regulates several genes involved in immune responses and is widely studied in immunity and cancer research. They show that the dynamics of NF-κB concentration are coupled to varying levels of a cytokin-like tumor necrosis factor (TNF). They show that when TNF oscillations have a low amplitude, NF-κB oscillations are periodic. We simulated periodic NF-κB oscillations using Heltberg and colleagues' Matlab code, available at https://github.com/Mathiasheltberg/ChaoticDynamicsInTranscriptionFactors.

*Chaotic NF-κB transcription.* Heltberg and colleagues[14] show that by increasing the amplitude of the TNF signal, the oscillating number of NF-κB molecules in their model becomes chaotic. We simulated chaotic NF-κB oscillations using Heltberg and colleagues' Matlab code, available at https://github.com/Mathiasheltberg/ChaoticDynamicsInTranscriptionFactors.

**Non-biological simulations.** Because there are only a limited number of biological simulations for which the presence of chaos has already been established, we also applied the Chaos Decision Tree Algorithm to a wide range of mathematical systems previously studied in the chaos theory and nonlinear time-series analysis literatures:

*Chaotic cubic map.* Venkatesan and Lakshmanan[50] describe a quasiperiodically forced cubic map, which can exhibit a large diversity of periodic, chaotic, and strange non-chaotic dynamics. In particular, the map exhibits many different routes to chaos. Their system is described by the following:

$$x_{i+1} = Q + f\cos(2\pi\theta_i) - Ax_i + x_i^3$$
$$\theta_{i+1} = \theta_i + \omega(\text{mod}\,1), \tag{8}$$

where $\omega = \frac{\sqrt{5}-1}{2}$ (the golden ratio). We set $f = -0.8$, $Q = 0$, and $A = 1.5$, which Venkatesan and Lakshmanan have shown lead to chaotic dynamics[50]. For the results reported in Table 2, we followed Dawes and Freeland[23] in taking a linear combination of **x** and **θ**: $\phi_i = x_i/6 + \theta_i/10$. Results for **x** individually are reported in Supplementary Table 15 (results for **θ** on its own are not informative, as **θ** is an independent, quasi-periodic process).

*Periodic cubic map.* Venkatesan and Lakshmanan[50] show that the system in Eq. (8) exhibits periodic (one-frequency torus) dynamics when $f = 0$, $Q = 0$, and $A = 1$. We picked these parameters for periodic dynamics. To get a time-series **φ** from the cubic map, we again took a linear combination of **x** and **θ**: $\phi_i = x_i/6 + \theta_i/10$. Results for **x** and **θ** individually are reported in Supplementary Table 15.

*Strange non-chaotic cubic map (HH).* We set $f = 0.7$, $Q = 0$, and $A = 1.88697$ for one type of strange non-chaotic dynamics, which Venkatesan and Lakshmanan[50] have shown bring the forced cubic map into a strange non-chaotic regime via the Heagy-Hammel route (i.e., collision of a period-doubled quasi-periodic torus with its unstable parent). Results for **x** individually are reported in Supplementary Table 15.

*Strange non-chaotic cubic map (S3).* We set $f = 0.35$, $Q = 0$, and $A = 0.35$ for a second type of strange non-chaotic dynamics, which Venkatesan and Lakshmanan[50] have shown push the forced cubic map into a strange non-chaotic regime via Type-3 Intermittency (i.e., inverse period-doubling bifurcation). Results for **x** individually are reported in Supplementary Table 15.

*Period-doubled cubic map.* Venkatesan and Lakshmanan[50] show that the system in Eq. (8) exhibits period-doubled dynamics when $f = -0.18$, $Q = 0$, and $A = 1.1$. We picked these parameters for period-doubled dynamics. To get a time-series **φ** from the cubic map, we again took a linear combination of **x** and **θ**: $\phi_i = x_i/6 + \theta_i/10$. Results for **x** individually are reported in Supplementary Table 15.

*Strange non-chaotic GOPY model.* The first known strange non-chaotic system was described by Grebogi, Ott, Pelikan, and Yorke, commonly referred to as the GOPY model[51]. The GOPY model is described by the following:

$$x_{i+1} = 2\sigma(\tanh x_i)\cos(2\pi\theta_i)$$
$$\theta_{i+1} = \theta_i + \omega(\text{mod}\,1), \tag{9}$$

where $\sigma = 1.5$, $\theta = 0.5$, and $\omega = \frac{\sqrt{5}-1}{2}$ (the golden ratio). To get a time-series **φ** from the GOPY model, we followed Dawes and Freeland[23] in taking a linear combination of **x** and **θ**: $\phi_i = x_i/6 + \theta_i/10$. Results for **x** individually are reported in Supplementary Table 15 (as is the case for the cubic map, results for **θ** on its own are not informative, as **θ** is an independent, quasi-periodic process).

*Chaotic logistic map.* The logistic map is one of the simplest known systems that can exhibit both periodic and chaotic behavior. It was originally introduced by biologist Robert May[52] as a discrete-time model of population growth. It is described by the following equation:

$$x_{i+1} = rx_i(1 - x_i) \tag{10}$$

where $x_i$ represents the ratio of the population size at time $i$ to the maximum possible population size. For chaotic dynamics[52], we set $r = 4$.

*Periodic logistic map.* For periodic dynamics[52] in the logistic map, we set $r = 3.5$ in Eq. (10).

**Chaotic Lorenz system.** Perhaps the most famous of all chaotic systems, the Lorenz model of atmospheric convection is described by the following system of equations[53]:

$$\frac{dx}{dt} = \sigma(y - x)$$
$$\frac{dy}{dt} = x(\rho - z) - y \tag{11}$$
$$\frac{dz}{dt} = xy - \beta z$$

where **x** is the rate of convection, **y** is the horizontal temperature variation, and **z** is the vertical temperature variation. Though the equations were initially meant to model atmospheric convection, identical equations have been found in models of a wide variety of physical systems, including lasers[79] and chemical reactions[80]. We set $\sigma = 10$, $\rho = 30$, and $\beta = \frac{8}{3}$, for which the Lorenz system exhibits chaos (determined by calculating the largest Lyapunov exponent of the system with these parameters, using Ramasubramanian's algorithm[81]). We integrated the equations for the Lorenz system using the Fourth Order Runge-Kutta method with an integration step of 0.01. To get a single time-series **ϕ** from the Lorenz model, we took a linear combination of **x** and **y**: $\boldsymbol{\phi} = \mathbf{x} + \mathbf{y}$. Results for **x**, **y**, and **z** individually are reported in Supplementary Table 15.

**Hyperchaotic generalized Henon map.** Data from hyperchaotic systems, which contain more than one positive Lyapunov exponent, can be difficult to distinguish from noise[25]. As such, hyperchaotic systems present a challenge to tests of determinism from time-series data, which might mistake hyperchaos for stochasticity. To demonstrate the robustness of the Chaos Decision Tree Algorithm's stochasticity test, we analyzed the Generalized Henon Map, which is described by the following equation:

$$x_{i+1} = a - x_{i-1}^2 - bx_{i-2} \tag{12}$$

We set $a = 1.76$ and $b = 0.1$, for which the Generalized Henon Map produces hyperchaos[54].

**Noise-driven sine map.** Freitas and colleagues[55] describe a non-chaotic, randomly driven system:

$$x_{i+1} = \mu \sin(x_i) + Y_i \eta_i \tag{13}$$

where $\mu = 2.4$, $Y_i$ is a random Bernoulli process that equals 1 with probability 0.01 and 0 with probability 0.99, and $\eta_i$ is a random variable uniformly distributed between $-2$ and 2. Freitas and colleagues show that a chaos-detection technique called "noise titration"[82] incorrectly classifies this system as chaotic.

**Freitas map.** Freitas and colleagues[55] also describe a nonlinear correlated noise process, which we here call the "Freitas map." The Freitas map contains dynamic noise added to a nonlinear moving average filter:

$$v_{i+1} = 3v_i + 4v_{i-1}(1 - v_i) \tag{14}$$

where $v_n$ is a uniform random variable distributed between 0 and 1. Freitas and colleagues show that the noise titration technique also incorrectly classifies this system as chaotic.

**Bounded random walk.** Nicolau[56] describes a bounded random walk (BRW), which is a globally stationary process with local unit roots (i.e. local non-stationarities):

$$X_t = X_{t-1} + e^k(e^{-\alpha_1(X_{t-1}-\tau)} - e^{\alpha_2(X_{t-1}-\tau)}) + \sigma_t \epsilon_t \tag{15}$$

where $\tau$, $k$, $\alpha_1$, $\alpha_2$, and $\sigma$ are parameters, and $\epsilon_t$ is a white noise error term. Note that the bounded random walk can be decomposed into a random walk, $X_t = X_{t-1} + \sigma_t \epsilon_t$, plus an adjustment function $e^k(e^{-\alpha_1(X_{t-1}-\tau)} - e^{\alpha_2(X_{t-1}-\tau)})$. The adjustment function serves to pull the random walk toward $\tau$ if the process deviates too far from $\tau$. Though it is a stationary process (albeit with local non-stationarities), the bounded random walk is often mis-classified as non-stationary by stationarity tests[83]. Following Nicolau[56] and Patterson[83], we set $\tau = 100$, $k = -15$, $\alpha_1 = 3$, $\alpha_2 = 3$, and $\sigma = 0.4$, which generates a random walk that remains roughly within the interval of $100 \pm 5$.

**Cyclostationary process.** A cyclostationary autoregressive process is essentially a combination of a noise-driven linear damped oscillator and linear relaxors. Cyclostationary systems are non-stationary because their probability distributions vary cyclically with time. Following Timmer[57], we simulate a cyclostationary process described by the following:

$$X_t = a_1 X_{t-1} + a_2 X_{t-2} + \epsilon_t \tag{16}$$

where $\epsilon_t$ is a white noise error term and

$$a_1 = 2\cos(2\pi/T)e^{-1/\tau}$$
$$a_2 = -e^{-2/\tau} \tag{17}$$

$\tau$ is the relaxation time and $T$ is the oscillation period. We set $\tau = 50$ and $T = 10$, which Timmer has shown leads to incorrect classification of this system as nonlinear or deterministic by surrogate tests that only use Fourier-based surrogates, which are strictly stationary.

**ARMA process.** A general autoregressive moving-average (ARMA) process is described by the following:

$$X_t = c + \epsilon_t + \sum_{i=1}^{p} \phi_i X_{t-i} + \sum_{i=1}^{q} \theta_i \epsilon_{t-i} \tag{18}$$

where $c$ is a constant, $\phi_1, ..., \phi_p$ and $\theta_1, ..., \theta_p$ are parameters, and $\epsilon_t, \epsilon_{t-1}, ...$ are white noise error terms. An ARMA process with a lag of 1, or an ARMA(1) process, is:

$$X_t = c + \epsilon_t + \phi X_{t-1} + \theta \epsilon_{t-1} \tag{19}$$

When $\phi < 1$, the ARMA process is (weakly) stationary. When $\phi$ is close to but less than 1 and $\theta \neq 0$, ARMA processes, though stationary, are often mis-classified as non-stationary by stationarity tests[84]. All ARMA processes simulated in this paper were lag 1, and we set $c = 0$ and $\phi = 0.99$. For the analyses in Supplementary Tables 1–4 and in Table 2, we drew $\boldsymbol{\theta}$ from a random, normal distribution with mean $\mu = 0$ and standard deviation $\sigma = 1$ for each simulation. We also tested ARMA(1) processes for $\boldsymbol{\theta}$ values fixed at $-0.5$, 0, 0.5, and 0.9 (Supplementary Table 19).

**Random walk.** A random walk is modeled by an autoregressive process with a unit root:

$$X_t = X_{t-1} + \epsilon_t \tag{20}$$

where $\epsilon$ is a white noise error term with mean $\mu = 0$ and standard deviation $\sigma = 1$. Random walks are non-stationary.

**Trended random walk.** A trended random walk introduces a secondary non-stationarity, namely, a linear trend, to the random walk:

$$X_t = X_{t-1} + b + \epsilon_t \tag{21}$$

where $\epsilon$ is a white noise error term with mean $\mu = 0$ and standard deviation $\sigma = 1$ and $b$ is the slope of the linear trend. For all trended random walks simulated in this paper, $b$ was randomly drawn from a Gaussian distribution with mean $\mu = 0$ and standard deviation $\sigma = 0.01$.

**Colored noise.** Colored noise refers to a stationary, stochastic process with a non-uniform power spectrum. White noise has a uniform power spectrum (meaning equal power at all frequencies); pink noise has a power spectral density proportional to $\frac{1}{f}$, where $f$ is frequency; red noise has a power spectral density proportional to $\frac{1}{f^2}$; blue noise has a power spectral density proportional to $f$; and violet noise has a power spectral density proportional to $f^2$. All colored noise signals were simulated using Zhivomirov's algorithm[58], available at https://www.mathworks.com/matlabcentral/fileexchange/42919-pink-red-blue-and-violet-noise-generation-with-matlab.

**Rössler system.** The Rössler system is described by the following system of differential equations[59]:

$$\frac{dx}{dt} = -wy - z$$
$$\frac{dy}{dt} = wx + ay \tag{22}$$
$$\frac{dz}{dt} = b + z(x - c)$$

We set $a = 0.2$, $b = 0.2$, and $c = 5.7$, for which the Rössler system exhibits chaos[59]. $w$ controls the frequency of the system's oscillations, and was set to 1. We integrated the equations for the Rössler system using the Fourth Order Runge-Kutta method with an integration step of 0.01. We generated 5,000,000 time-points, and then downsampled to 10,000 datapoints. To get a single time-series **ϕ** from the Rössler model, we took a linear combination of **x** and **y**: $\boldsymbol{\phi} = \mathbf{x} + \mathbf{y}$. Results for **x**, **y**, and **z** individually are reported in Supplementary Table 15.

**Ikeda map.** Ikeda and colleagues described a chaotic model of light passing through a nonlinear optical resonator[85]. The model can be simplified into a two-dimensional map[86]:

$$x_{i+1} = 1 + u(x_i \cos t_i - y_i \sin t_i)$$
$$y_{i+1} = u(x_i \sin t_i - y_i \cos t_i) \tag{23}$$

where $u$ is a parameter and

$$t_i = 0.4 - \frac{6}{1 + x_i^2 + y_i^2} \tag{24}$$

We set $u = 0.9$, for which the Ikeda map exhibits chaos[86]. Table 3 reports results for a linear combination of the two variables, $\boldsymbol{\phi} = \mathbf{x} + \mathbf{y}$. Results for **x** and **y** individually are reported in Supplementary Table 15.

**Hénon map.** The Hénon map[61] is a two-dimensional system of equations:

$$x_{i+1} = 1 - ax_i^2 + y_i$$
$$y_{i+1} = bx_i \tag{25}$$

We set $a = 1.25$ and $b = 0.3$, for which the Hénon map is periodic[87]. Table 3 reports results for a linear combination of the two variables, $\boldsymbol{\phi} = \mathbf{x} + \mathbf{y}$. Results for **x** and **y** individually are reported in Supplementary Table 15.

*Periodic Poincaré oscillator.* The Poincaré oscillator has been widely studied as a model of biological oscillations, particularly as a model of the effect of periodic stimulation on the dynamics of biological oscillators[88]. The oscillator is described by the following equations:

$$\frac{dr}{dt} = kr(1 - r)$$
$$\frac{d\Phi}{dt} = 2\Phi \tag{26}$$

where $k$ is a positive value that controls the oscillator's relaxation rate. The phase of this system is described by its angular coordinate $\phi$ in a unit cycle. Periodic stimulation of the system is modeled as a perturbation of magnitude $b$ away from the unit cycle, which leads to an instantaneous phase resetting of the phase of the oscillator, as determined by the following phase resetting curve:

$$g(\phi) = \frac{1}{2\pi} \arccos \frac{\cos 2\pi\phi + b}{\sqrt{1 + b^2 + 2b \cos 2\pi\phi}} (\text{mod } 1) \tag{27}$$

The period of the stimulation is determined by a parameter $\tau$. For periodic dynamics, we analyze the time-varying phase of the Poincaré oscillator with $b = 1.13$ and $\tau = 0.69$, which Guevara and Glass show leads to phase locking between the oscillator and the periodic perturbations[41].

*Quasi-periodic Poincaré oscillator.* For quasi-periodic dynamics[41] in the Poincaré oscillator, wet set $b = 0.95$ in Eq. (20), with an inter-stimulus delay $\tau = 0.75$.

*Chaotic Poincaré oscillator.* For chaotic dynamics[41] in the Poincaré oscillator, wet set $b = 1.13$ and $\tau = 0.65$.

*Stochastic Lorenz system.* To study the effect of dynamic noise on our algorithm's classification of stochastic chaotic systems, we took the Lorenz system described in Eq. (11), and added intrinsic/dynamic Gaussian noise to the **x** component of the system (we found that the system was far less sensitive to noise being injected into the **y** variable):

$$\frac{dx}{dt} = \sigma(y - x) + A\eta$$
$$\frac{dy}{dt} = x(\rho - z) - y \tag{28}$$
$$\frac{dz}{dt} = xy - \beta z$$

where $\eta_i$ is a normally distributed random variable with mean 0 and standard deviation 1, and $A$ is a parameter that controls the amplitude of the dynamic/intrinsic noise. As for the deterministic case, we set $\sigma = 10$, $\rho = 30$, and $\beta = \frac{8}{3}$. The stochastic Lorenz system was simulated using the Fourth Order Runge-Kutta method with an integration step of 0.01. Supplementary Table 18 reports results for different values of $A$, both for all system variables individually and for the linear combination $\mathbf{x} + \mathbf{y}$.

*Stochastic Rössler system.* We also took the Rössler system described in Eq. (22), and added dynamical Gaussian noise to the **x** component of the system:

$$\frac{dx}{dt} = \sigma(y - x) + A\eta$$
$$\frac{dy}{dt} = x(\rho - z) - y \tag{29}$$
$$\frac{dz}{dt} = xy - \beta z$$

where $\eta_i$ is a normally distributed random variable with mean 0 and standard deviation 1, and $A$ is a parameter that controls the amplitude of the dynamic/intrinsic noise. As for the deterministic case, we set $a = 0.2$, $b = 0.2$, and $c = 5.7$, and simulated the model using the Fourth Order Runge-Kutta method with an integration step of 0.01. We generated 5,000,000 time-points, and then downsampled to 10,000 datapoints. Supplementary Table 18 reports results for different values of $A$, both for all system variables individually and for the linear combination of $\mathbf{x} + \mathbf{y}$.

*Multivariate AR model.* We generated random multivariate autoregressive (AR) models using the Multivariate Granger Causality (MVGC) toolbox[89]. To create random regression matrices, we created random 5-node dense positive definite matrices using Matlab's `sprandsym.m` function, with a graph density of 1. To ensure stationary dynamics, we used the MVGC toolbox's `var_specrad.m` function to decay the coefficients of the random dense positive definite matrices so that their spectral radii were 0.8. To ensure uncorrelated noise in the resulting AR model, we created error matrices with diagonal elements set to 1 and off-diagonal elements set to 0. We then inputted these regression and error matrices into the MVGC toolbox's `var_to_tsdata.m` function to create multivariate time-series with 5 nodes and 10,000 time-points. We then applied the Chaos Decision Tree Algorithm to just the univariate activity of the first node of the resulting multivariate signal.

**Empirical data.** We here describe the real-world data analyzed in this paper, and how these data were previously classified as stochastic, periodic, or chaotic:

*A chaotic neuron integrated circuit.* Data recorded from an integrated circuit were kindly sent to us by Seiji Uenohara and colleagues. The circuit is a physical implementation of a chaotic neuron model that is based on the Hudgkin-Huxley equations[90]. The equations governing the circuit's behavior can be reduced to the following two-dimensional map:

$$\zeta(t + 1) = k_r\zeta(t) + af(\zeta(t) + b \cos(2\pi\theta(t))) + a$$
$$\theta(t + 1) = \theta(t) + \omega (\text{mod} 1) \tag{30}$$

where $f(\cdot)$ is a monotonically decreasing nonlinear output function, $b$ controls the amplitude of the quasi-periodic forcing, and $\omega = \frac{\sqrt{5}-1}{2}$ (the golden ratio). The quasi-periodic external forcing was inputted to the circuit using an analog board PXI-6289, which was also used to record the circuit's output. Varying the parameter $b$ can bring the circuit into periodic, strange non-chaotic, and chaotic states, which Uenohara and colleagues were able to classify by analyzing the consistency of the circuit's response to an external input[27]. There are 10 datasets recorded from the circuit's chaotic state.

*A strange non-chaotic neuron integrated circuit.* There are 10 datasets recorded from the strange non-chaotic state of Uenohara and colleagues' circuit.

*A periodic neuron integrated circuit.* There are 10 datasets recorded from the periodic state of Uenohara and colleagues' circuit.

*Chaotic laser.* Hübner and colleagues[28] used phase portrait, correlation integral, and autocorrelation function analyses to detect chaos in the intensity pulsing of an unidirectional far-infrared NH3 ring laser. Laser data were downloaded from https://www.pdx.edu/biomedical-signal-processing-lab/chaotic-time-series.

*Stellar flux of a strange non-chaotic star.* We analyzed stellar flux from KIC 5520878, the only known non-artificial strange non-chaotic system[29]. Data were sent to us by John F. Lindner, who, together with colleagues, determined the status of KIC 5520878 as a strange non-chaotic system using a series of spectral scaling analyses[29]. Data were originally obtained from the Mikulski Archive for Space Telescopes. Because there are large shifts in the data due to the stellar flux being recorded in different pixels of the Kepler Space Telescope, we visually inspected the data to find a relatively stable period (i.e. a period in between large shifts) and then detrended the data. We thus exclusively analyzed time points 11,620 to 14,003 from the dataset analzed in Lindner and colleagues' paper[29].

*North Atlantic Oscillation Index.* We analyzed the monthly mean North Atlantic Oscillation (NAO) Index from January 1950 to December 2018. The NAO Index is the difference in atmospheric pressure at sea level between the Azores high and the Icelandic low, and has been shown by several groups of researchers, employing a range of techniques, to be stochastic[25,91–96]. Data were downloaded from the Climate Prediction Center website (http://www.cpc.ncep.noaa.gov/).

*Parkinson's tremor.* We analyzed recordings of a Parkinson's patient's hand acceleration, measured for 30 s at a sampling rate of 1000 Hz by piezoresistive accelerometers. Through analyses of correlation integrals, Poincaré and return maps, Lyapunov exponents, and the $\delta$-$\epsilon$ method, Timmer and colleagues[57] showed that this Parkinson's tremor was a nonlinear stochastic oscillator. Data were downloaded from http://jeti.uni-freiburg.de/path_tremor/readme.

*Essential tremor.* We analyzed recordings of hand acceleration from a patient with an essential tremor, also measured for 30 seconds at a sampling rate of 1000 Hz by piezoresistive accelerometers. As they did with the Parkinson's tremor, Timmer and colleagues used correlation integrals, Poincaré and return maps, Lyapunov exponents, and the $\delta$-$\epsilon$ method to show that this essential tremor was a nonlinear stochastic oscillator. Data were downloaded from http://jeti.uni-freiburg.de/path_tremor/readme.

*Heart rate (healthy subjects).* Five heart beat (RR-interval) time-series recordings from healthy subjects were downloaded from Physionet[31]: https://www.physionet.org/challenge/chaos/. The signals were recorded using continuous ambulatory (Holter) electrocardiograms, and are in sinus rhythm. Outliers were filtered out of the data using Physionet's WFDB software package. Though a full 24 h of data were available for each subject, we only took the first 2.78 h of data, corresponding to 10,000 time-points. This was both to save on computation time and to be consistent with the length of other time-series analyzed in this paper.

*Heart rate (congestive heart failure patients).* Five heart beat (RR-interval) time-series recordings from patients with congestive heart failure were downloaded from Physionet[31]. Like the healthy rate signals, these data were recorded using continuous ambulatory (Holter) electrocardiograms, are in sinus rhythm, and were filtered for outliers. Though a full 24 h of data were available for each subject, we only took the first 2.78 h of data.

*Heart rate (atrial fibrillation).* Five heart beat (RR-interval) time-series recordings from patients with congestive heart failure were downloaded from Physionet[31]. Like the healthy rate signals, these data were recorded using continuous ambulatory (Holter) electrocardiograms and were filtered for outliers, but are not in sinus rhythm. We only took the first 2.78 h of data, corresponding to 10,000 time-points.

**Parameters and largest Lyapunov exponents for data in Table 5.** We here describe the methods used to generate data with different degrees of chaos for the analyses reported in Table 5, as well as the methods used to calculate largest Lyapunov exponents in these systems.

*Logistic map.* The logistic map has only a single parameter, $r$ (see above). Following Bandt and Pompe[32], we varied $r$ between 3.5 and 4, in intervals of 0.001,

to generate 501 10,000 time-point signals with different levels of chaos. Ground-truth largest Lyapunov exponents were calculated using the derivative method, which does not involve generating time-series data.

*Hénon map.* To generate different degrees of chaos in the Hénon map, we varied its *a* parameter (see above) between 1 and 1.4, in intervals of 0.001, to generate 401 10,000 time-point signals with different degrees of chaos. Ground-truth largest Lyapunov exponents were calculated using code provided in *Dynamical Systems with Applications using Matlab*[97], available at https://github.com/springer-math/Dynamical-Systems-with-Applications-using-MATLAB/.

*Lorenz system.* For the Lorenz system, we varied its $\sigma$ parameter between 5.75 and 15, in intervals of 0.05, and generated 10,000 time-points per simulation. Within this parameter range, the Lorenz system is chaotic, but displays varying degrees of chaos. To calculate largest Lyapunov exponents for each parameter, we used the algorithm provided by Ramasubramanian[81], which, like the algorithms used for the logistic and Hénon maps, does not involve generating time-series data.

*Mean-field cortical model.* Following Steyn-Ross, Steyn-Ross, and Sleigh[20], different levels of chaos in their mean-field cortical model were generated by varying two parameters: postsynaptic inhibitory response and inhibitory diffusion. The postsynaptic inhibitory response parameter ($\lambda_i$ in their model) was varied between 0.98 and 1.018 in intervals of 0.001, and the inhibitory diffusion parameter ($D_2$ in their model) was varied between 0.1 and 0.8 in intervals of 0.05, producing a total of 585 parameter configurations. We simulated 25,000 time-points in the model, with no downsampling, so we could again test the effect of the Chaos Decision Tree Algorithm's automated downsampling on permutation entropy' ability to track level of chaos in continuous systems. Unfortunately, there are no tools for analytically estimating the largest Lyapunov exponent of the mean-field cortical model, and so largest Lyapunov exponents were approximated by running two noise-free simulations of the model for each parameter configuration, with very slightly different initial conditions, and fitting a line to the rate of divergence between the two simulations from the beginning of the simulations to the point when their divergence rate saturates and flattens out. The slope of the fitted line is taken as the estimate of the largest Lyapunov exponent[10]. While Steyn-Ross, Steyn-Ross, and Sleigh's code for estimating largest Lyapunov exponents using this method requires subjective evaluation of where to fit the line (i.e. finding the non-saturated part of the divergence rate plot), we automated this process by fitting a line from the beginning of the divergence rate plot to the point where its mean abruptly changes, reflecting saturation; this point was determined using Matlab's findchangepts.m function. Simulations with approximated largest Lyapunov exponents less than −5 or greater than 5 were excluded from the analysis, as these are likely poor estimates; visual inspection confirmed that for these cases, there was often no clear point of saturation for the divergence rate between simulations, and so lines were often automatically fitted to particularly steep, short sub-segments of the plot. Visual inspection further confirmed that in most other cases, there was a clear, linear rate of divergence between simulations followed by saturation, and that the automatically fitted line was a good fit.

*Neuron integrated circuit.* The integrated circuit data analyzed in Table 5 are the same as those analyzed in Table 4. Because the circuit is a physical implementation of a known and simple two-dimensional system of equations, Uenuhara and colleagues used those equations to calculate the ground-truth largest Lyapunov exponents of the circuit in its three different states (periodic, strange non-chaotic, and chaotic), and report those largest Lyapunov exponents in their paper[27].

**Statistics and reproducibility.** In reporting the performance of the Chaos Decision Tree Algorithm, we only made recourse to statistical tests in Table 5 and Supplementary Table 14, where we report the (two-tailed) *p*-values of Spearman correlations between largest Lyapunov exponents and permutation entropies. Because these correlations were calculated against the same set of ground-truth largest Lyapunov exponents for each system, *p*-values were Bonferroni-corrected for multiple comparisons. Elsewhere, we only report the fraction out of all datasets that were correctly classified as stochastic, periodic, or chaotic (with exact sample sizes provided in all tables). Within the Chaos Decision Tree Algorithm itself, statistical tests appear in two locations. First, if the user chooses to test for stationarity, then the pipeline will only proceed to test for stochasticity if the test signal passes a (two-tailed) stationarity test with $\alpha = 0.05$. Moreover, the pipeline uses surrogate statistics to test for determinism: if the permutation entropy of the inputted signal lies outside the distribution of permutation entropies of 1000 surrogates (which is equivalent to a two-tailed statistical test with $\alpha = 9.99e{-}4$), then the data are classified as being generated by a predominantly deterministic system. No sample size calculation was performed before analyzing the data presented, as our results simply report classification accuracy for a large number of datasets (and no statistical analyses were performed beyond reporting classification accuracy). No data were excluded from the analysis. Data in Tables 1, 2 were used to optimize our algorithm, which was then re-tested on datasets in Tables 3, 4. Finally, note that no *p*-value is associated with the *K*-statistic outputted by the 0–1 test for chaos. To facilitate the reproducibility of our analyses, we have included code alongside our pipeline, as well as links to code in our Methods, for generating all simulated datasets tested in this paper. Links to most empirical datasets analyzed here have been provided in the Methods.

**Reporting summary.** Further information on research design is available in the Nature Research Reporting Summary linked to this article.

## Data availability

Links to Matlab scripts for simulating the mean-field cortical model and NF-κB transcription are provided in the Methods. Matlab scripts for simulating all other systems described in this paper are provided along with the code for the Chaos Decision Tree Algorithm at https://figshare.com/s/80891dfb34c6ee9c8b34 (DOI: doi.org/10.6084/m9.figshare.7476362.v7). All empirical datasets except for the integrated circuit recordings are freely available online, and URLs to each data source are provided in the Methods.

## Code availability

The code for the Chaos Decision Tree Algorithm is provided at https://figshare.com/s/80891dfb34c6ee9c8b34 (DOI: doi.org/10.6084/m9.figshare.7476362.v7).

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

## Acknowledgements

We thank Seiji Uenohara and colleagues for kindly sending us data recorded from their chaotic neuron integrated circuit, and we thank John F. Lindner for sending us data on the stellar flux of KIC 5520878. We also thank Michael Jansson for his feedback on stationarity testing.

## Author contributions

D.T. conceived of the project, wrote the code for the Chaos Decision Tree Algorithm, analyzed results, and wrote the manuscript. F.T.S. and M.D. contributed to discussing, editing, and revising the paper.

## Competing interests

The authors declare no competing interests.
