## [Peer Review File · Communications Biology]

Reviewers' comments:

Reviewer #1 (Remarks to the Author):

This is a very interesting and well written paper. It deals with detecting chaotic dynamics from data, in particular with the aim of being able to confirm if a series of experimental measurements show chaos or not. The paper presents a kind of toolbox for this task. And very nicely the toolbox is freely available online.

I have some comments I should like the authors to consider.

1. The authors state that their method can distinguish stochastic, periodic and chaotic signals. But there are also quasiperiodic and period doubled signals. Can the method detect those also ?

2. The procedure as a whole is explained in the quite nice Fig. 1. Nevertheless, I would say it appears a little too much as a 'black box' (maybe even somewhat 'magic'). Some of the included methods I should really like to be explained more. See next point.

3. There is a quite large box explaining various important concepts. This is surely fine. However, there are some almost trivial concepts like linear and non-linear. Then I miss both Dicky-Fuller test and Schreiber algorithm in this box. Now that Kolmogorov-Sinai is so well explained in the box, I am really missing the other two concepts.

4. Two dimensional dynamical systems are discussed quite a lot. However, I am missing a more general discussion about dimensions. Experimental signals are most often one-dimensional but can also be of higher dimensions. Does the algorithm work for all dimensions. And in this connection: people often use time delays in data analysis. Is such a technique useful or not useful here ?

Altogether this paper adds an interesting contribution to the extremely important question of how to distinguish between chaotic and stochastic signals. And in particular for experimental data. I am not sure this paper 'completely' solves this fundamental issue but it is a nice addition to the discussions.

In my opinion the paper can be published in Communications Biology after modifications.

Reviewer #2 (Remarks to the Author):

This article proposes a test for deterministic chaos in theoretical models and in experimental data. The authors developed an "automated processing pipeline" that they say is available online. I have not attempted to access (I am not sure where it is) the processing tools. The authors claim a very high level of accuracy on a number of different data sets in the presence of noise. They note some problems but do not analyze why the algorithms might break down in particular cases.

I read this article from the perspective of the long history of algorithms being proposed to detect chaos, strong claims that various naturally occurring data sets are chaotic, and eventual recognition of faults of the techniques being assessed, e.g. see Theiler J, Rapp PE. Re-examination of the evidence for low-dimensional, nonlinear structure in the human electroencephalogram. *Electroencephalography and clinical Neurophysiology*. 1996 Mar 1;98(3):213-22 or Kanters JK, HOLSTEIN-RATHLOU NH, Agner E. Lack of evidence for low-dimensional chaos in heart rate variability. *Journal of cardiovascular*

electrophysiology. 1994 Jul;5(7):591-601.

In biology, there are also experimental examples that I find convincing in which chaos has been demonstrated. The examples I am most familiar with involve periodic stimulation of biological oscillators for systems in which good models exist, and dynamics can be compared over a range of parameters, e.g. Guevara MR, Glass L, Shrier A. Phase locking, period-doubling bifurcations, and irregular dynamics in periodically stimulated cardiac cells. *Science*. 1981 Dec 18;214(4527):1350-3; Hayashi H, Nakao M, Hirakawa K. Chaos in the self-sustained oscillation of an excitable biological membrane under sinusoidal stimulation. *Physics Letters A*. 1982 Mar 15;88(5):265-6; Kaplan DT, Clay JR, Manning T, Glass L, Guevara MR, Shrier A. Subthreshold dynamics in periodically stimulated squid giant axons. *Physical Review Letters*. 1996 May 20;76(21):4074.

The above paragraph has a caveat though. In any real system there is necessarily noise (e.g. due to random channel opening and closing). When I say that there is chaos, I mean that the deterministic chaotic aspect of the dynamics is still clear despite the noise, and the dynamics are similar to what is found in noiseless models.

Along with attempts to identify chaos in naturally occurring data, there have been a variety of claims for the utility of chaos in various physiological processes. In the words of R. Pool in the 1989 article: *Is it healthy to be chaotic?*. *Science*. 1989 Feb 3;243(4891):604-7, "Chaos may provide a healthy flexibility for the heart, brain, and other parts of the body." An early discussion of possible neural systems that might display chaos was Guevara MR, Glass L, Mackey MC, Shrier A. Chaos in neurobiology. *IEEE Transactions on Systems, Man, and Cybernetics*. 1983 Sep(5):790-8. An early proponent of the importance of chaos in neurophysiology was the late W. J. Freeman. The authors mention several other new hypothetical ways in which chaos may be useful for biological function. Despite claims for the role of chaos in biological systems, to the best of my knowledge, there are not clear examples showing the role of chaos in normal functioning. There is a pretty big literature discussing these matters.

Now 20-30 years after the papers mentioned above were published, there is perhaps a resurgence of interest in chaos. It is possible that the current generation will be able to go much more deeply than was possible before. However, I think that the current paper should do a better job of indicating that the basic premise of the current work is not new, and give enough reference to the literature that an interested reader would be able to find out easily what was done before and be forewarned about possible problems. Many of the mathematical aspects of the test are reviewed in "The 0-1 Test for Chaos: A Review", by Georg A. Gottwald and Ian Melbourne, in Ch. Skokos et al. (eds.), *Chaos Detection and Predictability*, Chaos detection and predictability, pages 221-247 (2016).

The current paper should have some conceptual explanation of what the various steps are in the decision tree. I understand that many of the steps may be very technical, but some more explanation of why they may work is needed. I would have liked to see time traces going from original data and showing each step: denoising, downsampling, etc.

The current paper carries out analyses of a large number of models and data sets, and in this way claims the general utility of the method. But the discussion of each case is superficial.

Here is an issue arising from one example cited above (Guevara et al., 1981). Take data in which a cardiac oscillator is periodically stimulated. For some frequencies and amplitudes of stimulation there appears to be chaos. The collected data can consist of membrane voltage showing action potentials with the superimposed stimulus artifact, or just the phases in the cardiac cycle of the stimulus artifact. To apply the 0-1 test could one use either the phases or the membrane voltage? Would one need the

same length of data for the both cases, and how much data would that be. Would 10-20 stimuli be enough? How frequently do I need to sample the data? (the upstroke of the action potential is fast, as is the stimulus artifact.) Maybe there is some insight into this in the article, but if so, I missed it.

Here are some technical aspects that could be better explained.

1. When noise is added, it could be (i) added to the dynamical equations or (ii) added after the dynamics computed without noise. Case (i) would be important for real systems. I think this was only looked at for a couple of examples (#10 and #16). Could this be made clearer?
2. What is the definition of stationarity? Are data sets that show $1/f$ noise like heart rate variability stationary?
3. All equations need to be checked. Differential equations (like the Rossler equation, Lorenz equation, Mackey-Glass equation) need to be represented as differential equations, not as difference equations. Fix the formula for the Freitas map.
4. You give the integration step for the Rossler and Lorenz equations, but not the Mackey-Glass equation.
5. For the Rossler equation you took a linear combination of x and y . Does the method work if you only use x , y , or z ? Do you need the same amounts of data for each case? Letellier has pointed out that some variables have richer dynamics than others in these equations, Letellier C, Aguirre LA. Investigating nonlinear dynamics from time series: The influence of symmetries and the choice of observables. *Chaos: An Interdisciplinary Journal of Nonlinear Science*. 2002 Sep 20;12(3):549-58.
6. There needs to be some description of what skills and knowledge, if any, would be needed to run and appropriately assess the data. I would have thought that this would be stated clearly in the article, and perhaps it was, but if so, I missed it.
7. The limits of the current tests are not stated explicitly.
8. How does the algorithm work on very high dimensional attractors in deterministic systems with aperiodic chaotic dynamics?
9. The definition of chaos, must also state that the dynamics are bounded. Exponential growth would be chaotic under the definition given here.

In the past, there have been huge disagreements about whether some data sets, such as normal heart rate variability, are chaotic (for example, see L. Glass. Introduction to controversial topics in nonlinear science: Is the normal heart rate chaotic? *Chaos* **19**, 028501 (2009) and several other articles in the September 2009 issue of *Chaos*). It would have been useful if the authors had examined some data sets of this sort to see how their algorithm would deal with such data. In general, many biological systems contain elements of high dimensional dynamical systems (lots of variables and time delays), fluctuating environments and intrinsic noise in the elements of the system. Although it would have been possible and also fascinating if such data had shown the presence of a low dimensional chaotic dynamics, claims of low dimensional chaos have often been based on inappropriate application of subtle algorithms.

To date, investigating the question of whether such data sets are chaotic or not has not been very fruitful. Of course, this would change if someone could give a clear demonstration, backed by data and

theory, of a low dimensional chaotic system that was important for some critical biological process or indicative of pathology.

Leon Glass

Reviewers' comments:

Reviewer #1 (Remarks to the Author):

This is a very interesting and well written paper. It deals with detecting chaotic dynamics from data, in particular with the aim of being able to confirm if a series of experimental measurements show chaos or not. The paper presents a kind of toolbox for this task. And very nicely the toolbox is freely available online.

I have some comments I should like the authors to consider.

1. The authors state that their method can distinguish stochastic, periodic and chaotic signals. But there are also quasiperiodic and period doubled signals. Can the method detect those also ?

We thank Reviewer #1 for pointing out our omission of these two dynamical states. We applied our algorithm to both the quasi-periodic Poincaré oscillator and the period-doubled cubic map, and found that it performed with perfect accuracy for both systems, which we have now included in our revised paper (Table 3).

2. The procedure as a whole is explained in the quite nice Fig. 1. Nevertheless, I would say it appears a little too much as a 'black box' (maybe even somewhat 'magic'). Some of the included methods I should really like to be explained more. See next point.

3. There is a quite large box explaining various important concepts. This is surely fine. However, there are some almost trivial concepts like linear and non-linear. Then I miss both Dicky-Fuller test and Schreiber algorithm in this box. Now that Kolmogorov-Sinai is so well explained in the box, I am really missing the other two concepts.

In light of this comment, we have added a more formal mathematical description of the 0-1 test to the Online Methods, and also substantially increased the number of concepts explained in Box 1. These include: measurement noise, dynamic noise, quasiperiodicity, period-doubling, the Augmented Dicky-Fuller Test, Schreiber de-noising, and surrogate testing.

4. Two dimensional dynamical systems are discussed quite a lot. However, I am missing a more general discussion about dimensions. Experimental signals are most often one-dimensional but can also be of higher dimensions. Does the algorithm work for all dimensions. And in this connection: people often uses time delays in data analysis. Is such a technique useful or not useful here ?

In the revised manuscript, we clarified the dimensionality of the various systems analyzed in this paper. These include one-dimensional maps (the logistic map and the generalized Henon map), the one-dimensional Mackey-Glass time delay differential equation, a number of two-dimensional systems, a five-dimensional model of NF-kB transcription, and a model of cortical dynamics consisting of 120 interacting fourteen-dimensional systems. We did not observe any changes in accuracy as a function of system dimensionality.

Regarding the reviewer's second point, there is no direct use for time delays for the chaos-detection portion of the algorithm, as the 0-1 test for chaos does not involve any time delays.

But, time delays do come into play for both the calculation of permutation entropy and for the Schreiber de-noising algorithm. However, for the sake of simplicity, we used a time delay of 1 for both cases.

Altogether this paper adds an interesting contribution to the extremely important question of how to distinguish between chaotic and stochastic signals. And in particular for experimental data. I am not sure this paper 'completely' solves this fundamental issues but it is a nice addition to the discussions.

In my opinion the paper can be published in Communications Biology after modifications.

We thank Reviewer #1 for their helpful and encouraging comments, and hope that we have satisfactorily addressed their concerns.

Reviewer #2 (Remarks to the Author):

This article proposes a test for deterministic chaos in theoretical models and in experimental data. The authors developed an "automated processing pipeline" that they say is available online. I have not attempted to access (I am not sure where it is) the processing tools. The authors claim a very high level of accuracy on a number of different data sets in the presence of noise. They note some problems but do not analyze why the algorithms might break down in particular cases.

I read this article from the perspective of the long history of algorithms being proposed to detect chaos, strong claims that various naturally occurring data sets are chaotic, and eventual recognition of faults of the techniques being assessed, e.g. see Theiler J, Rapp PE. Re-examination of the evidence for low-dimensional, nonlinear structure in the human electroencephalogram. *Electroencephalography and clinical Neurophysiology*. 1996 Mar 1;98(3):213-22 or Kanters JK, HOLSTEIN-RATHLOU NH, Agner E. Lack of evidence for low-dimensional chaos in heart rate variability. *Journal of cardiovascular electrophysiology*. 1994 Jul;5(7):591-601.

In biology, there are also experimental examples that I find convincing in which chaos has been demonstrated. The examples I am most familiar with involve periodic stimulation of biological oscillators for systems in which good models exists, and dynamics can be compared over a range of parameters, e.g. Guevara MR, Glass L, Shrier A. Phase locking, period-doubling bifurcations, and irregular dynamics in periodically stimulated cardiac cells. *Science*. 1981 Dec 18;214(4527):1350-3; Hayashi H, Nakao M, Hirakawa K. Chaos in the self-sustained oscillation of an excitable biological membrane under sinusoidal stimulation. *Physics Letters A*. 1982 Mar 15;88(5):265-6; Kaplan DT, Clay JR, Manning T, Glass L, Guevara MR, Shrier A. Subthreshold dynamics in periodically stimulated squid giant axons. *Physical Review Letters*. 1996 May 20;76(21):4074.

The above paragraph has a caveat though. In any real system there is necessarily noise (e.g. due to random channel opening and closing). When I say that there is chaos, I mean that the deterministic chaotic aspect of the dynamics is still clear despite the noise, and the dynamics are similar to what is found in noiseless models.

Along with attempts to identify chaos in naturally occurring data, there have been a variety of claims for the utility of chaos in various physiological processes. In the words of R. Pool in the 1989 article: Is it healthy to be chaotic?. *Science*. 1989 Feb 3;243(4891): 604-7, "Chaos may provide a healthy flexibility for the heart, brain, and other parts of the body." An early discussion of possible neural systems that might display chaos was Guevara MR, Glass L, Mackey MC, Shrier A. Chaos in neurobiology. *IEEE Transactions on Systems, Man, and Cybernetics*. 1983 Sep(5):790-8. An early proponent of the importance of chaos in neurophysiology was the late W. J. Freeman. The authors mention several other new hypothetical ways in which chaos may be useful for biological function. Despite claims for the role of chaos in biological systems, to the best of my knowledge, there are not clear examples showing the role of chaos in normal functioning. There is a pretty big literature discussing these matters.

Now 20-30 years after the papers mentioned above were published, there is perhaps a resurgence of interest in chaos. It is possible that the current generation will be able to go much more deeply than was possible before. However, I think that the current paper should do a better job of indicating that the basic premise of the current work is not new, and give enough reference to the literature that an interested reader would be able to find out easily what was done before and be forewarned about possible problems. Many of the mathematical aspects of the test are reviewed in "The 0-1 Test for Chaos: A Review", by Georg A. Gottwald and Ian Melbourne, in Ch. Skokos et al. (eds.), *Chaos Detection and Predictability, Chaos detection and predictability*, pages 221-247 (2016).

The current paper should have some conceptual explanation of what the various steps are in the decision tree. I understand that many of the steps may be very technical, but some more explanation of why they may work is needed. I would have liked to see time traces going from original data and showing each step: denoising, downsampling, etc.

We thank Reviewer #2 for their insightful feedback, and for pointing us to important papers in the history of the study of chaos in biological systems. In light of these comments, we have incorporated these citations into our revised paper to more adequately situate our work within its historical context. We have also clarified the differences between measurement vs. intrinsic noise at several points in the paper. Finally, to further clarify each step of the algorithm, we have added additional technical explanations to both the Online Methods and Box 1, and included sample time traces illustrating each step of the algorithm (Supplementary Figure 3).

The current paper carries out analyses of a large number of models and data sets, and in this way claims the general utility of the method. But the discussion of each case is superficial.

In our revised paper, we have now incorporated additional details into the descriptions of most of the test systems described in our Online Methods.

Here is an issue arising from one example cited above (Guevara et al., 1981). Take data in which a cardiac oscillator is periodically stimulated. For some frequencies and amplitudes of stimulation there appears to be chaos. The collected data can consist of membrane voltage showing action potentials with the superimposed stimulus artifact, or just the phases in the cardiac cycle of the stimulus artifact. To apply the 0-1 test could one use either the phases or the membrane voltage? Would one need the same length of

data for the both cases, and how much data would that be. Would 10-20 stimuli be enough? How frequently do I need to sample the data? (the upstroke of the action potential is fast, as is the stimulus artifact.) Maybe there is some insight into this in the article, but if so, I missed it.

It is difficult to answer this question directly without the empirical data in question, though we thank both the editor and Reviewer #2 for their efforts to find these data for us. Instead, we followed Reviewer #2's subsequent recommendation (made to the editor via email) that we could apply our algorithm to the periodically stimulated Poincaré phase oscillator instead. We applied the algorithm to a time-series of successive phases of the oscillator in chaotic, periodic, and quasi-periodic states (as assessed by Guevara and Glass, 1982). As with other simulations in our paper, we generated 10,000 time-points for each of these states, for which our algorithm performed perfectly. These new results are now presented in our revised paper.

The reviewer's second comment, regarding the length of data needed for accurate results, was very helpful, and led to a number of new analyses in our paper. First, we realized that the optimal cutoff for classifying a system as periodic or chaotic is a function of time-series length: the longer the time-series, the higher the optimal cutoff (Supplementary Figure 5). In light of this new finding, we modified our algorithm to pick an optimal cutoff based on the length of the inputted time-series. To assess how this affects the algorithm's performance for short time-series, we re-ran our analyses on sub-samples of the datasets in Tables 1-3. We found that the algorithm still performed at very high accuracy for signals with just 1,000 time-points and 5,000 time-points, though performance for some systems (notably periodic NF-kB transcription and the quasi-periodic Poincaré oscillator) did decrease with less data. We thus re-emphasize in the discussion that, as a rule of thumb, the longer the inputted time-series, the more accurate the algorithm will be.

Finally, in light of the comment made here about different system observables, as well as the related comment below about different observables of the Rössler system, we re-applied our algorithm to individual variables of all multi-dimensional systems analyzed in this paper (Supplementary Table 15), with the exception of the cortical model and NF-kB transcription, for which the observables of interest are biologically well-defined. We found that the choice of observables did not significantly affect the performance of the algorithm.

Here are some technical aspects that could be better explained.

1. When noise is added, it could be (i) added to the dynamical equations or (ii) added after the dynamics computed without noise. Case (i) would be important for real systems. I think this was only looked at for a couple of examples (#10 and #16). Could this be made clearer?

This is a very important point, which we now explore in more detail in our revised paper. First, we more clearly define measurement noise (i.e. noise that is added after the dynamics have been simulated) vs. dynamic/intrinsic noise (i.e. noise added to a system's dynamical equations). These definitions are in Box 1. We also now demonstrate (Supplementary Table 16) that higher levels of intrinsic noise are more likely to lead to classification of a system as stochastic, which is to be expected.

2. What is the definition of stationarity? Are data sets that show 1/f noise like heart rate variability stationary?

In Box 1, we define a stationary process as one whose joint probability distribution is time-invariant. In other words, for a stationary process, statistical properties like mean and variance do not fluctuate over time. The power spectrum of a signal does not necessarily reflect its stationarity or lack thereof. For example, a random walk is nonstationary while red noise is stationary, though both have roughly the same power spectrum. The Dickey-Fuller stationarity test used by our algorithm can distinguish these two cases (Table 2, Supplementary Figure 4).

3. All equations need to be checked. Differential equations (like the Rossler equation, Lorenz equation, Mackey-Glass equation) need to be represented as differential equations, not as difference equations. Fix the formula for the Freitas map.

All equations and their corresponding code were checked thoroughly, and updated. While doing so, we caught two other mistakes in our previous submission: the parameters used for the stochastic Lorenz system were incorrect, and the time-series for the GOPY model used only the theta variable, as opposed to a linear combination of the two variables (as we had written in the Online Methods). These have been corrected in the revised paper, and we re-ran all analyses involving these two systems.

4. You give the integration step for the Rossler and Lorenz equations, but not the Mackey-Glass equation.

We now clarify that the Mackey-Glass equation was integrated with a first-order Euler method with an integration step of 1.

5. For the Rossler equation you took a linear combination of x and y. Does the method work if you only use x, y, or z? Do you need the same amounts of data for each case? Letellier has pointed out that some variables have richer dynamics than others in these equations, Letellier C, Aguirre LA. Investigating nonlinear dynamics from time series: The influence of symmetries and the choice of observables. Chaos: An Interdisciplinary Journal of Nonlinear Science. 2002 Sep 20;12(3):549-58.

This was a very important point, and we thank the reviewer for bringing this to our attention. We had previously examined linear combinations of variables of our multi-dimensional systems under the assumption that in real-life situations, recorded data will usually contain features of several system variables (as you point out above with respect to the periodically stimulated cardiac oscillators). That said, we re-ran our analyses on individual variables (Supplementary Table 15), and found that classification accuracy was not markedly reduced for individual variables. That said, we now point to Letellier's paper in the revised Discussion.

6. There needs to be some description of what skills and knowledge, if any, would be needed to run and appropriately assess the data. I would have thought that this would be stated clearly in the article, and perhaps it was, but if so, I missed it.

We now clarify that to use our method, a researcher only needs to provide a time-series recording. In other words, the method does not require extensive knowledge of nonlinear time-

series analysis. That said, it helps to understand what the algorithm is doing; we hope that the new details that have been added to the Online Methods and Box 1 will help in that regard.

7. The limits of the current tests are not stated explicitly.

We now devote a paragraph in the revised Discussion section to the limitations/shortcomings of the tests presented in this paper.

8. How does the algorithm work on very high dimensional attractors in deterministic systems with aperiodic chaotic dynamics?

We applied our algorithm to one high dimensional system, which is the mean-field cortical model of Steyn-Ross et al. The model consists of 120 connected 14-dimensional oscillators. Our algorithm was applied to just one variable from one oscillator; that variable was the mean firing rate of one of 120 populations of excitatory neurons in the model (which corresponds to the real-life EEG, MEG, or ECoG signals that neuroscientists would analyze). We now explicitly describe the dimensionality of this system, as well as other systems, in our Online Methods.

9. The definition of chaos, must also state that the dynamics are bounded. Exponential growth would be chaotic under the definition given here.

Thank you for pointing out this omission. We have amended our definition of chaos in both the introduction and Box 1 to clarify that chaotic systems are bounded.

In the past, there have been huge disagreements about whether some data sets, such as normal heart rate variability, are chaotic (for example, see L. Glass. Introduction to controversial topics in nonlinear science: Is the normal heart rate chaotic? Chaos {\bf 19**}, 028501 (2009) and several other articles in the September 2009 issue of Chaos). It would have been useful if the authors had examined some data sets of this sort to see how their algorithm would deal with such data. In general, many biological systems contain elements of high dimensional dynamical systems (lots of variables and time delays), fluctuating environments and intrinsic noise in the elements of the system. Although it would have been possible and also fascinating if such data had shown the presence of a low dimensional chaotic dynamics, claims of low dimensional chaos have often been based on inappropriate application of subtle algorithms.**

Although we had initially envisioned this paper as a methodological proof of concept (with plans to apply our algorithm to more empirical data, in particular to neural recordings, in future papers), we agree that it would be interesting to do a first-pass at an empirical application and examine heart rate signals. Thus, we downloaded the 15 outlier-filtered heart rate recordings from Physionet's "Is the Normal Heart Rate Chaotic?" challenge. In agreement with simulation-based arguments that these signals reflect a nonlinear stochastic process, and that past classifications of chaos in these data are erroneous, our method consistently classified these signals as stochastic (Table 7). We have now presented these new results in our revised paper.

To date, investigating the question of whether such data sets are chaotic or not has not been very fruitful. Of course, this would change if someone could give a clear demonstration, backed by data and theory, of a low dimensional chaotic system that was important for some critical biological process or indicative of pathology.

We agree that simply asking whether or not a given biological process is chaotic is not as important as demonstrating that chaos is functionally relevant to some biological processes. We have tried to emphasize this important point more clearly in our revised paper, and hope that our method will motivate other researchers, who might not have considered this question in the biological system they are studying, to elucidate the role of chaos in both biological function and dysfunction.

Reviewers' comments:

Reviewer #1 (Remarks to the Author):

The paper has been sufficiently well revised to warrant publication.

Reviewer #3 (Remarks to the Author):

The authors assemble several existing tools for classifying an input time series into whether it is nonstationary or stationary, and in the latter case, whether it is stochastic or deterministic and further classify a found deterministic series into chaos or regular. They have made specific choices of tools for each classification step. The classification accuracy of their proposed entire procedure must then be tied to the appropriateness and the power of the tools chosen by them.

For instance, they propose to use the augmented Dickey Fuller (ADF) test for deciding whether a time series is stationary. It is, however, well known that the ADF test may fail in many cases, especially when the underlying process has a moving-average component; see Chapters 6 and 9 of Patterson, K. (2011), *Unit Root Tests in Time Series*, Vol. I. Moreover, the ADF test is also known to have problems in discerning nonlinear stationary processes with local unit roots, from nonstationary processes; see vol. 2 of Patterson's book.

In the authors' approach for discerning deterministic processes from stationary, stochastic processes, they make use of the surrogate method. Section 4.4 in the book entitled "Chaos: A Statistical Perspective" by Chan and Tong (2011) discusses certain conditions for the validity of the surrogate method, and emphasizes the importance to transform the data to normality before applying the surrogate method. The paper under review contains no discussion regarding the need for normality transformation.

Chan and Tong (2011) provides a comprehensive exposition of the then state of art research on studying the sensitivity to initial values for chaotic system with dynamic noise. It seems that the authors' method will likely classify such systems to be stationary stochastic processes, in the case of non-negligible dynamic noise. None of the simulations in the paper include such models; it may be interesting to check whether the proposed method breaks down with increasing dynamic noise. If, indeed, the proposed method tends to classify such systems to be stochastic and non-chaotic, such a classification will be misleading.

My main problem with the paper is that through limited simulation and real data analysis, they put forward their assemblage classifier as a universal classifier for general use, especially for biological science. The specific choices of tools in the proposed method have its own limitations of applicability which are swept under the carpet, by making it a one size fit all, universally applicable method. It will be more useful if the assemblage allows various options in each of its sub-classification steps, so a user can make appropriate choices for analyzing his/her data.

Some technical comments:

In Table 2: The classification results are sometime unstable. For instance, the Lorenz system was correctly classified for noise levels 0%, 10%, 20% and 40%, but the classification accuracy is 27/100 for noise level 30%. Why? Are the random number seed the same for each noise level? If not, it will introduce another source of variation due to using different seeds. Even so, it is hard to understand the big discrepancy. Also, for the colored noises, there are no results for noise level higher than 0%. Why?

There are several typos in Box 1. Please check.

The authors assemble several existing tools for classifying an input time series into whether it is nonstationary or stationary, and in the latter case, whether it is stochastic or deterministic and further classify a found deterministic series into chaos or regular. They have made specific choices of tools for each classification step. The classification accuracy of their proposed entire procedure must then be tied to the appropriateness and the power of the tools chosen by them.

For instance, they propose to use the augmented Dickey Fuller (ADF) test for deciding whether a time series is stationary. It is, however, well known that the ADF test may fail in many cases, especially when the underlying process has a moving-average component; see Chapters 6 and 9 of Patterson, K. (2011), Unit Root Tests in Time Series, Vol. I. Moreover, the ADF test is also known to have problems in discerning nonlinear stationary processes with local unit roots, from nonstationary processes; see vol. 2 of Patterson's book.

We thank the reviewer for pointing out this shortcoming of the ADF test, and for drawing our attention to Patterson's books. His books helped us considerably in this revision. In light of these comments, as well as the analyses reported in Patterson's books, we added a number of test systems to our paper: bounded random walks (which are globally stationary and locally non-stationary), a cyclostationary autoregressive process, an autoregressive process with a moving average component, random walks, and random walks with linear trends. Based on Patterson's analyses, we also included two additional non-parametric unit root tests: Lo and MacKinlay's variance ratio test and Breitung's variance ratio test. In agreement with Patterson, we found that Breitung's variance ratio test was more robust to these new edge cases.

In the authors' approach for discerning deterministic processes from stationary, stochastic processes, they make use of the surrogate method. Section 4.4 in the book entitled "Chaos: A Statistical Perspective" by Chan and Tong (2011) discusses certain conditions for the validity of the surrogate method, and emphasizes the importance to transform the data to normality before applying the surrogate method. The paper under review contains no discussion regarding the need for normality transformation.

While we have not seen normality transformations mentioned in more recent reviews of surrogate testing (e.g. Lancaster et al, 2018), we followed up on this comment by systematically testing whether a normality transformation using the Box-Cox method, as recommended by Chan and Tong (2011), improved the performance of surrogate-based stochasticity tests. We did not find any improvement following normality transformations (Supplementary Table 2).

As mentioned in the cover letter, we also did not find any improvement following the exclusion of signals classified as non-stationary by Breitung's variance ratio test. In fact, we found that our stochasticity test performed well for all non-stationary signals tested, as well as for signals with local unit roots and moving average components (Supplementary Table 4). As such, we have now taken out the stationarity test component of our pipeline. That said, we have modified our code such that a user can decide to include any of the stationarity tests we analyzed, as well as to normality-transform their data before generating surrogate signals, using any of the surrogate algorithms tested.

Chan and Tong (2011) provides a comprehensive exposition of the then state of art research on studying the sensitivity to initial values for chaotic system with dynamic

noise. It seems that the authors' method will likely classify such systems to be stationary stochastic processes, in the case of non-negligible dynamic noise. None of the simulations in the paper include such models; it may be interesting to check whether the proposed method breaks down with increasing dynamic noise. If, indeed, the proposed method tends to classify such systems to be stochastic and non-chaotic, such a classification will be misleading.

We note that our previous submission did include this analysis, per the request of Reviewer #2. That analysis is in Supplementary Table 18. We did find that, as expected, classifications of stochasticity for both the stochastic Lorenz and stochastic Rössler systems became more frequent as levels of dynamic noise were increased. However, we do not think that this classification is misleading, as these systems might reasonably be considered as nonlinear and stochastic rather than chaotic, in the case of non-negligible dynamic noise. That said, we agree with the argument in Chan and Tong (2011) that there is no mathematically rigorous dividing line between “operationally” stochastic and “operationally” deterministic systems as a function of dynamic noise level, i.e., that it is unclear how small a level of dynamic noise is “small enough.” We also note that the test of “operational” determinism described in Chan and Tong is somewhat subjective, as it entails deciding whether the bandwidth h of a local linear regression is sufficiently close to zero (leaving open the “practical issue of deciding ‘how small is small’”). Chan and Tong address this issue of subjectivity using a bootstrap method, but that bootstrap method is itself not a proper statistical test. Surrogate-based tests, on the other hand, are statistically well-defined. As such, our surrogate-based approach may be seen as an alternative test of operational determinism. Moreover, the classification of stochasticity in the stochastic Lorenz and Rössler systems for high levels of dynamic noise may be seen as evidence that these systems are, as we think is intuitively true, operationally stochastic.

My main problem with the paper is that through limited simulation and real data analysis, they put forward their assemblage classifier as a universal classifier for general use, especially for biological science. The specific choices of tools in the proposed method have its own limitations of applicability which are swept under the carpet, by making it a one size fit all, universally applicable method. It will be more useful if the assemblage allows various options in each of its sub-classification steps, so a user can make appropriate choices for analyzing his/her data.

We have modified our code such that users can select among any of the alternative subroutines tested in our paper, and we emphasize this point throughout our manuscript. We also now include additional emphasis in our paper that our method is not guaranteed to yield accurate results, and that it should be used in conjunction with accurate computer/mathematical models of the system at hand, when possible.

Some technical comments:

In Table 2: The classification results are sometime unstable. For instance, the Lorenz system was correctly classified for noise levels 0%, 10%, 20% and 40%, but the classification accuracy is 27/100 for noise level 30%. Why? Are the random number seed the same for each noise level? If not, it will introduce another source of variation due to using different seeds. Even so, it is hard to understand the big discrepancy. Also, for the colored noises, there are no results for noise level higher than 0%. Why?

We caught a bug in the Runge-Kutta algorithm that was used to integrate the Lorenz and Rössler equations, and so we re-ran all analyses in the paper involving either of these two systems. This discrepancy for 30% noise has now disappeared, and otherwise our results or conclusions have not changed in any meaningful way. That said, we note that there was a similar, though less dramatic discrepancy for the bounded random walk in Table 2 (which was classified as stochastic with near-perfect accuracy for all noise levels except 20%, for which only 59/100 cases were classified as stochastic). Because this discrepancy is not systematic, it is unclear why it should appear for just this one system (although we note that a majority of simulations of the bounded random walk with 20% noise were still correctly classified as stochastic). Random number seeds and initial values were different for every simulation in this paper, regardless of noise level.

For the colored noise, we did not add additional white noise on top of the noise signals, as doing so would flatten or “whiten” their power spectra. We now clarify this point in the paper.

There are several typos in Box 1. Please check.

We hope that we have caught and corrected all typos.

Reviewers' comments:

Reviewer #3 (Remarks to the Author):

The revision is a significant improvement, and I appreciate the authors to enhance their algorithm to allow more user inputs on the make-up of the algorithm.

Two comments:

(i) The authors misunderstands the ARMA model which is incorrectly specified. The current noise term is not included on the RHS of (18); the correct formulation should have that term included. For instance, the ARMA(1,1) model should be (in latex notation)

$$X_t = c + \phi X_{t-1} + \epsilon_t + \theta \epsilon_{t-1}$$

However, in the revision, this model is written as

$$X_t = c + \phi X_{t-1} + \theta \epsilon_{t-1}$$

which is essentially an AR(1) model.

Thus, the authors simulated data from AR(1) not ARMA(1,1) model!

Moreover, the MA coefficient θ should be set to some fixed values, for instance -0.5, 0, 0.5, 0.9, and not randomly drawn from some normal distribution, so it can shed light on the effects of the strength of the MA component on the proposed test. In particular, the authors may need to re-do this part of the analysis.

(ii) I am not convinced that simulation results reported support the authors' claim that normality transformation is inconsequential. A stronger stress test is needed. For instance, take an exponential transformation on the data and re-run the experiments to compare the various surrogate methods.

The revision is a significant improvement, and I appreciate the authors to enhance their algorithm to allow more user inputs on the make-up of the algorithm.

We thank the reviewer for this encouraging comment, and for their prior feedback.

Two comments:

(i) The authors misunderstands the ARMA model which is incorrectly specified. The current noise term is not included on the RHS of (18); the correct formulation should have that term included.

For instance, the ARMA(1,1) model should be (in latex notation)

$$X_t = c + \phi X_{t-1} + \epsilon_t + \theta \epsilon_{t-1}$$

However, in the revision, this model is written as

$$X_t = c + \phi X_{t-1} + \theta \epsilon_{t-1}$$

which is essentially an AR(1) model.

Thus, the authors simulated data from AR(1) not ARMA(1,1) model!

This was actually a typo in our Online Methods. The Matlab code that we used to simulate the ARMA(1,1) model was correct - i.e., it included the error term that was accidentally omitted from the equation in the Methods section.

Moreover, the MA coefficient θ should be set to some fixed values, for instance -0.5, 0, 0.5, 0.9, and not randomly drawn from some normal distribution, so it can shed light on the effects of the strength of the MA component on the proposed test. In particular, the authors may need to re-do this part of the analysis.

In light of this comment, we further analyzed the performance of our algorithm for the MA coefficients -0.5, 0, 0.5, and 0.9, while keeping $\phi=0.99$. We found that the algorithm performed well for all MA coefficient values (Supplementary Table 19).

(ii) I am not convinced that simulation results reported support the authors' claim that normality transformation is inconsequential. A stronger stress test is needed. For instance, take an exponential transformation on the data and re-run the experiments to compare the various surrogate methods.

To address this, we re-ran all analyses in Supplementary Table 2 using a rank-based inverse normal transformation, which yields an exactly Gaussian distribution. Performance using this robust normality transformation was similar to performance after the Box-Cox transformation, and was worse than performance with no normality transformation (Supplementary Table 2). We therefore maintain no transformation as the default setting of our algorithm, though we have modified our code such that users can choose to transform their signal with either the Box-Cox method or the rank-based inverse normal transformation method.